# Position: Causality Is Key for Interpretability Claims to Generalise

**Shruti Joshi** [1]  **Aaron Mueller** [2]  **David Klindt** [3]  **Wieland Brendel** [4]  **Patrik Reizinger** [* 4]  **Dhanya Sridhar** [* 1]

## Abstract

Interpretability research on large language models (LLMs) has yielded important insights into model behaviour, yet recurring pitfalls persist: findings that do not generalise, and causal interpretations that outrun the evidence. Our position is that causal inference specifies what constitutes a valid mapping from model activations to invariant high-level structures, the data or assumptions needed to achieve it, and the inferences it can support. Specifically, Pearl's causal hierarchy clarifies what an interpretability study can justify. Observations establish associations between model behaviour and internal components. Interventions (e.g., ablations or activation patching) support claims how these edits affect a behavioural metric (e.g., average change in token probabilities) over a set of prompts. However, counterfactual claims—i.e., asking what the model ouput would have been for the same prompt under an unobserved intervention—remain largely unverifiable without controlled supervision. We show how causal representation learning (CRL) operationalises this hierarchy, specifying which variables are recoverable from activations and under what assumptions. Together, these motivate a diagnostic framework that helps practitioners select methods and evaluations matching claims to evidence such that findings generalise.

## 1. Introduction

Interpretability research on LLMs has produced a growing toolkit for linking model behaviour to internal structure, e.g. circuits trace task computations (Olah et al., 2020; Elhage et al., 2021; Wang et al., 2023), activation patching local-ises contributions of specific model components (Vig et al., 2020), sparse autoencoders (SAEs) surface human-readable features (Cunningham et al., 2023). Yet practitioners frequently encounter a gap between local success and reliable deployment, e.g., a linear probe achieves high accuracy, but steering the same direction fails to produce reliable behavioural change (Tan et al., 2024), or an ablation suppresses a behaviour on the test set, yet the same circuit proves brittle under distribution shift (Miller et al., 2024). These patterns raise a general epistemological challenge: high predictive accuracy does not, by itself, establish the existence of manipulable mechanisms (Kambhampati, 2024).

To move beyond purely correlational accounts, recent interpretability work has focused on grounding model behaviour in internal structure. This shift has been framed as a turn toward *mechanistic* interpretability. In its original usage, mechanistic explanation characterised how causes produce effects (Machamer et al., 2000; Woodward, 2002; 2003), asking not just *what* a system computes but *how* (Marr & Poggio, 1979), and which events count as *the* cause(s) (Halpern & Pearl, 2005a;b). When interpretability researchers set out to "reverse engineer the detailed computations performed by transformers" (Elhage et al., 2021), they implicitly inherited these causal commitments. Claims that a circuit *computes* a function, that an attention head *mediates* a behaviour, or that an SAE feature *controls* a capability are claims about causal influence within the model. Causal inference provides vocabulary needed to make these claims precise: What variables are we positing? What interventions define the causal relationships? What alternative explanations remain compatible with our evidence?

These questions have well-developed answers in causal inference. To formalize the variables and claims, we can specify an *estimand*, the precise quantity a method targets. For instance, a probe targets decodability of a concept (an associational estimand), which is distinct from asking how the output changes if the decoded concept is perturbed (an intervention-effect estimand). An *intervention class* then specifies the manipulations that would be needed to estimate the intervention-effect estimand, i.e., the change in a chosen output metric induced by perturbing the representation of the decoded concept. To obtain a unique estimate, we must determine which perturbations are indistinguishable given our measurements, which is captured by an *equivalence*

---

[*]Equal advising; authors listed in alphabetical order. [1]Mila - Québec AI Institute & Université de Montréal [2]Boston University [3]Cold Spring Harbor Laboratory [4]Max-Planck-Institute for Intelligent Systems, ELLIS Institute Tübingen, University of Tübingen. Correspondence to: Shruti Joshi <shrutijoshi98@gmail.com>.

*Proceedings of the 43rd International Conference on Machine Learning*, Seoul, South Korea. PMLR 306, 2026. Copyright 2026 by the author(s).

*class*. Finally, we must ask when conclusions drawn from these measurements are transferable, a question addressed by causal transportability (Pearl, 2009; Bareinboim & Pearl, 2012).

Without this scaffolding, interpretability claims become ambiguous, where the same word—"mechanism", "feature", "circuit"—can refer to different estimands and intervention classes, making results hard to compare across papers (Saphra & Wiegreffe, 2024; Mueller, 2024). More practically, methods can succeed locally yet fail when deployed because the evidence answers a different question than the one implied by the claim (Bereska & Gavves, 2024; Sharkey et al., 2025). For instance, decodability (an association) may be used to justify control (an intervention effect). Importantly, the interpretability community has itself identified many of these challenges: recent works advocate for necessity and sufficiency testing (Heimersheim & Nanda, 2024), document how entanglement limits single-feature interventions (Mueller et al., 2025), and call for more rigorous evaluation methodology (Makelov et al., 2024; Sharkey et al., 2025). Our contribution is not to discover these issues but to provide a unified causal framework that makes them precise. We use Pearl's causal hierarchy (Pearl, 2009) to diagnose potential claim-evidence mismatches, and causal representation learning (CRL) to specify the assumptions required for common claims in interpretability research.

One response to these limitations is the emerging shift to *pragmatic interpretability* (Nanda et al., 2025), which advocates for iterating against measurable proxies rather than "using model internals to understand or explain behaviour". This posture is not new (Bas, 1980; Dewey, 1948; Chang, 2004; Potochnik, 2017) and has precursors in explainable machine learning (Nyrup & Robinson, 2022). where interpretability is treated as purpose-relative and evaluated by downstream use rather than discovering a fixed set of concepts (Doshi-Velez & Kim, 2017; Lipton, 2018; Miller, 2019). We take this shift as a useful point of comparison and argue that causal inference provides complementary tools for specifying when proxy-based success should generalise, and when it may not (Craver, 2007; Jacovi & Goldberg, 2020; Leeb et al., 2025).

**Our position is that interpretability claims should be stated in the language of *causal inference and identifiability*: specify the estimand, the intervention class, and the equivalence class implied by the available evidence.** The payoff is **practical**: this framing helps practitioners choose methods that actually answer the question of interest, diagnose mismatch failures between method and goal, and state the conditions under which the conclusions will transfer. A shared causal vocabulary also helps verify and compare claims across methods and applications.

**Conflict of Interest Disclosure.** The authors declare no financial conflicts of interest.

## 2. Position: Interpretability Requires Identifiable Causal Quantities

Interpretability research aims to make precise claims about model internals, yet—as in any application of the scientific method—such claims require committing to a well-defined target of inference, which often remains implicit in questions like "What does this head do?" or "Is this the honesty feature?" Without a precise target, even well-designed metrics risk validating structure that is not meaningfully different from random baselines (Heap et al., 2025; Méloux et al., 2025). Notably, this is a known problem when testing many hypotheses in high-dimensional data (Bennett et al., 2009), not one unique to interpretability. We argue that making interpretability claims more reliable requires three steps, formalised with the language of causality, for which we briefly introduce informal meanings of key terms:

> **ESTIMANDS AND IDENTIFIABILITY**
>
> **Estimand:** A quantity that would answer the question of interest, if it can be computed exactly.
>
> **Estimator:** A procedure approximating the estimand from data.
>
> **Equivalence class:** Hypotheses that are indistinguishable given interactions with the model.
>
> **Identifiability:** An estimand is identifiable up to an equivalence class if it is constant within the class but varies across classes such that estimating the estimand determines the class, but not the hypothesis within it.

**The causality recipe.** First, we use the causal ladder (see box below) to provide a taxonomy for distinguishing associational, interventional, or counterfactual questions (§ 2.1). This makes the target question explicit and mathematically precise. Second, we characterise the sufficient evidence and assumptions to identify the answer to the causal question (§ 2.2). Third, we ask which of that evidence is actually accessible, and how (§ 2.3).

> **PEARL'S CAUSAL LADDER**
>
> 👁 **L1 · Associational.** Statistics from observed data. *Query:* Does $A$ correlate with $B$?
>
> ⇄ **L2 · Interventional.** Effects of controlled modifications. *Query:* Does changing $A$ change $B$?
>
> ↯ **L3 · Counterfactual.** Alternative outcome for the same instance under an unobserved intervention. *Query:* For this input, would changing $A$ have changed $B$?

Different rungs answer different questions. Choosing the right one depends on your goal. Higher rungs require stronger evidence: L($k$) evidence does not license L($k$+1) claims (Pearl, 2009; Bareinboim et al., 2022) but L($k$+1) evidence licences L($k$) claims.

**Notation.** Details are in § A. Consider a pretrained model with $L$ layers and parameters $\theta$. For input $\mathbf{x} \in \mathbb{R}^{d_x}$, define layerwise activations recursively as $\mathbf{a}^{(0)} := \mathbf{x}$ and $\mathbf{a}^{(l)} := f_\theta^{(l)}(\mathbf{a}^{(l-1)})$ for $l = 1, \ldots, L$, where $\mathbf{a}^{(l)} \in \mathbb{R}^{d_a^{(l)}}$ denotes the raw activations at layer $l$. The model output is given by $\mathbf{y} := \mathbf{a}^{(L)}$. We distinguish activations from representations $\mathbf{h}^{(l)} := \phi^{(l)}(\mathbf{a}^{(l)})$, where $\phi^{(l)} : \mathbb{R}^{d_a^{(l)}} \to \mathbb{R}^{d_h^{(l)}}$ is a (possibly learned) map [1] to a basis where structure may be more apparent. A feature is simply a subspace $S \subset \mathbb{R}^{d_h^{(l)}}$.[2] Whether a feature is interpretable or meaningful is an empirical question requiring additional evidence.

## 2.1. Interpretability Questions Are Causal Questions

We argue that the questions of interpretability research can be mapped to the three rungs of Pearl's causal ladder and we use this classification to align each claim with the evidence required to support it.

This perspective clarifies a recurring pattern in the literature (Arditi et al., 2024; Bills et al., 2023; Bricken et al., 2023; Rajamanoharan et al., 2024): purely associational evidence (L1) is often reported in the language of counterfactual or interventional claims (L3/L2), as illustrated in Tab. 1. The resulting rung mismatch has concrete consequences for reliability and safety, as interventions that appear effective on benchmark evaluations frequently fail under distribution shift. We illustrate the implications at each rung using a running *example of refusal*, expressed in terms of conditional probabilities and potential outcomes.

---

WORKED EXAMPLE: CAUSING REFUSAL

👁 **L1** Do certain activations correlate with refusal behaviour? Given prompts $\mathbf{x}$ labeled by whether the model refuses ($\mathbf{y} = 1$) or complies ($\mathbf{y} = 0$), an associational query asks whether a representation $\mathbf{h}$ is predictive of refusal under the observational distribution: $p(\mathbf{y} \mid \mathbf{h}) \neq p(\mathbf{y})$.

✔ *Licenses:* Refusal is decodable from this layer.

✖ *Does not license:* The model represents refusal here, or this layer computes refusal.

⇵ **L2** Can we induce refusal by manipulating activations? Let $\mathrm{do}(\mathbf{h} := \widetilde{\mathbf{h}})$ denote an intervention that replaces the representation at a chosen

---

[1] Since our arguments apply to any fixed layer, we drop the layer index and write $\mathbf{a}, \mathbf{h}, \phi$.

[2] See § A.2 for other usages of the term.

layer with a fixed value $\widetilde{\mathbf{h}}$, leaving the remainder of the computation intact. An interventional query compares $p(\mathbf{y} \mid \mathrm{do}(\mathbf{h} := \widetilde{\mathbf{h}}))$ against the baseline $p(\mathbf{y})$ or against alternative interventions.

✔ *Licenses:* Intervening here changes refusal under tested conditions.

✖ *Does not license:* This is *the* refusal mechanism, or that it generalises to novel jailbreaks.

🗲 **L3** For a specific prompt on which the model was jailbroken, what activation change *would have* caused refusal? Given an observed triple $(\mathbf{x}_0, \mathbf{h}_0, \mathbf{y}_0)$ where $\mathbf{y}_0$ represents harmful compliance, a counterfactual query seeks $\widetilde{\mathbf{h}}$ such that $\mathbf{y}_{\mathbf{h} \leftarrow \widetilde{\mathbf{h}}}(\mathbf{x}_0)$ corresponds to refusal.

✔ *Licenses:* For this forward pass, the intervention would have caused refusal.

✖ *Does not license:* Generalises to other inputs without a structural model.

---

A comprehensive reference with the full ladder taxonomy and worked examples is provided in § D; Tab. 1 illustrates the mapping for common methods. The ladder specifies the question's rung, and the estimand specifies the quantity needed to answer the question of interest.

A core concern, to which we now turn, is *identifiability*: whether the available evidence is sufficient to uniquely determine the chosen estimand. For instance, observing that $A$ correlates with $B$ is consistent with $A$ causing $B$, but equally consistent with a confounder causing both. Interventional evidence can distinguish these cases; associational evidence cannot. Thus, Pearl's causal ladder classifies causal questions by the type of evidence (associational, interventional, counterfactual) required to answer them, but it does not ensure that the evidence pins down a unique answer.

## 2.2. The Right Question Is Not Enough. Answers Must Be Identifiable

After specifying an estimand, we must establish its identifiability, i.e., whether the available evidence rules out alternative values of the estimand. Identifiability is therefore best understood as a statement about *what structure or quantity is invariantly recoverable, not about what entities exist*. It characterises an equivalence class of explanations consistent with the evidence, not a unique ground truth (see § B.1 for a philosophical discussion).

Consider the interventional (L2) estimand $p(\mathbf{y} \mid \mathrm{do}(\mathbf{h}^{(l)} := \widetilde{\mathbf{h}}^{(l)}))$ that measures whether the activations at a layer causally influence a model's rate of refusal beyond some baseline rate $p(\mathbf{y})$. Suppose we only observe that $\mathbf{h}^{(l)}$ accurately predicts refusal. The estimand is not identifiable from the evidence we have, i.e., the available evidence could be equally well explained by earlier layers encoding an al-

*Table 1.* Interpretability methods typically produce **associational or interventional evidence (L1–L2), yet the interpretations we'd like to draw often implicitly require counterfactual reasoning (L3)**. Recognising this rung gap clarifies what additional evidence is needed to justify stronger claims.

| METHOD | IDEAL GOAL | | WHAT THE EVIDENCE SUPPORTS | |
|---|---|---|---|---|
| SPARSE AUTOENCODERS | ♘L3 | The learned features correspond to a unique set of concepts. *Identifiability claim.* | ◉L1 | A sparse basis that minimises reconstruction error on the training data. *Describes a basis, without yet establishing uniqueness.* |
| AUTO-INTERPRETABILITY EXPLANATIONS | ♘L3 | This feature corresponds to the underlying concept named by the description. *Semantic assignment.* | ◉L1 | The description predicts when the feature activates on held-out text. *Distinguishes activating from non-activating contexts, without confirming the feature's causal role.* |
| CIRCUIT DISCOVERY | ♘L3 | This circuit is a key mediator of the behaviour. *Causal attribution.* | ⇶L2 | Ablating this circuit changes model behaviour on evaluated prompts. *An intervention effect for the chosen ablation, not yet a unique localisation.* |

ternate concept that affects both the activations $\mathbf{h}^{(l)}$ and refusal.

Thus, locating an estimand on the causal ladder helps determine the strength of the evidence needed for identifiability. However, there are practical considerations for identifiability. For example, interventional evidence (L2) is costly: perturbing neurons and measuring behavioural changes requires many experiments and human annotation of outputs with interpretable concepts. Such practical considerations have paved the way for unsupervised methods like sparse autoencoders (SAEs), which learn a change of basis $\mathbf{h}^{(l)} := \phi^{(l)}(\mathbf{a}^{(l)})$ regularised toward sparsity, with the hypothesis that sparse coordinates are interpretable. However, even if SAE features are interpretable, that implies neither uniqueness nor identifiability. The use of either auto-interpretability explanations (L1 evidence; Bills et al. 2023; Paulo et al. 2024) or interventions (L2 evidence) cannot pin down the meaning of a feature precisely, which requires L3 evidence (as seen in § 2.1). In fact, auto-interp scores can attain high values even on random structure (Heap et al., 2025). These problems, we argue, can be characterised and understood with an identifiability lens. Namely, identifiability results prove that without additional structure, unsupervised recovery of latents is impossible (Hyvärinen & Pajunen, 1999; Locatello et al., 2019). The central question, then, is not whether unsupervised methods can discover structure, but *under what conditions the structure they recover is causally meaningful*.

**Identifiability in CRL.** The field of causal representation learning (CRL, Schölkopf et al. 2021) focuses on identifiable learning of causal variables. Concretely, CRL generally assumes that some ground-truth human-interpretable latent variables $\mathbf{c} \in \mathbb{R}^{d_c}$ produce observations $\mathbf{x}$ via an unknown generative process $g : \mathbb{R}^{d_c} \to \mathbb{R}^{d_x}$. The underlying variables $\mathbf{c}$ are considered to be causal variables since we can imagine intervening on them as humans. The goal of CRL is to develop assumptions about the observations or generative function $g$ to render the unmixing function $\mathbf{c} = g^{-1}(\mathbf{x})$ identifiable up to a simple equivalence class (e.g. permutation and rescaling of $\mathbf{c}$). Typically, identifiability requires observations gathered under multiple interventions on the latents $\mathbf{c}$ to learn both the latent factors and their causal structure.

**Applying CRL to pretrained LLM activations.** CRL methods offer a way to learn the map $\mathbf{h}^{(l)} := \phi^{(l)}(\mathbf{a}^{(l)})$ to reinterpret activations $\mathbf{a}^{(l)}$ in a coordinate system where the axes have causal semantics. Indeed, recent papers (Joshi et al., 2025; Geiger et al., 2025; Goyal et al., 2025; Marconato et al., 2023; Rajendran et al., 2024; Song et al., 2025) have already begun developing sufficient assumptions to identify some concept classes from observations reflecting natural concept variation. The practical question is whether the assumptions under which identifiability is guaranteed hold when deployed, and whether the resulting equivalence class suffices for the task at hand. While making these assumptions explicit and testing design choices against them may require extra effort upfront, they provide a practical payoff: identifiability guarantees can reduce experimentation costs by specifying how a claim is supported, independent of model size, data scale, or compute.

**Concrete Example.** Sparse shift autoencoders (SSAEs) Joshi et al. (2025) leverage paired samples that reflect diverse perturbations to an unknown number of concepts. The identifiability result guarantees the varied latents observed in data are identified up to permutations and scaling. If SSAEs are trained on data that reflecting changes to human

notions of target concepts (e.g., sentiment), such concepts are provably recovered. However, if the naturally occurring perturbations always jointly affect, e.g., sentiment and topic, the identified quantities will not reflect sentiment or topic separately. This interpretation of CRL results can be linked to the idea of affordances, which we describe next.

---

CONCEPTS AS AFFORDANCES

While classical CRL assumes ground-truth latents **c** and a ground-truth generative process $g$, interpretability lacks true concept labels. Therefore, we ask not whether a representation recovers true latents, but what it *affords* for specific interactions.
**Affordance:** What an interpreter can discover about a model depends on the interactions available to them (Gibson, 1979). Different probing methods—like different tools—reveal different structure. This may include novel structure or mechanisms, potentially requiring neologisms to describe.
**Example:** Consider two iron filings—one magnetised, and one not. They are indistinguishable under interactions such as: looking, weighing, or picking up. Only interaction with a magnet makes the distinction observable. Likewise, a representation may encode structure that only becomes visible under the right *interaction*.

---

**Identifiability as Interaction-Relative.** We cannot test whether sentiment varies independently of topic unless we observe settings where one changes without the other. What can be identified is therefore bounded by the variations the system affords. As in first-contact[3], behaviour alone admits multiple interpretations—"cold" may reflect temperature, discomfort, or a metaphor—so evidence identifies only an equivalence class consistent with the interactions performed (Quine, 1960; Davidson, 1973). Interpretation is thus bidirectional (Ayonrinde, 2025), and may surface structure for which humans lack vocabulary, requiring the creation of new words (neologisms) (Schut et al., 2025; Hewitt et al., 2025). Anthropomorphic pattern-seeking (Buckner, 2019; Marks et al., 2025b) risks projecting human concepts onto "alien" structure, as seen in topic modeling where labels impose meaning on arbitrary clusters (Chang et al., 2009).

---

WHY REQUIRE IDENTIFICATION GUARANTEES IF A METHOD WORKS IN PRACTICE?

There's remarkable evidence showing that interpreted concepts can improve chess grandmasters (Schut et al., 2025), steering vectors can induce refusal (Arditi et al., 2024), and probes can accurately predict sentiment (Conneau et al., 2018) or truthfulness (Marks & Tegmark,

---

2024). Then one might ask: why do we need to recover latent variables that correspond to a uniquely identifiable or causally meaningful structure?
This is because the evidence is not guaranteed to generalise. Model edits revert on paraphrases (Hoelscher-Obermaier et al., 2023); steering vectors miscalibrate OOD (Turner et al., 2023); probes fail under distribution shift (Lovering et al., 2021). This might suffice for exploration. But we know that for safety-critical deployment, asking why something works, and when it will stop working is unavoidable. Identifiability makes this question answerable: it specifies *which* assumption to verify (e.g. concept variation) and *what* failure to expect if it breaks. This provides means to rule out failure modes pre-deployment, rather than post-hoc.

---

**Relocating supervision.** There is no free lunch in interpretability: supervision enters both through the affordances a method is designed to leverage (rather than labelled datasets), and through downstream interpretation; unsupervised objectives and evaluations do not, by themselves, distinguish representations that uncover meaningful variables from those that do not. Thus, without identifiability, fitting a latent variable model recovers some latent coordinates that reproduce the observed distribution, such that any invertible remixing of these coordinates (such as, a rotation, or even a nonlinear reparameterisation) yields the same solution under the designed objective [4]. Consequently, assigning individual semantics (such as, "latent $i$ is speech", and "latent $j$ is music") is not stable: another equally good solution can blend speech and music across multiple latent components. Identifiability pins down this meaning, as an invariance across the equivalence class of observationally equivalent solutions, i.e., if the latent coordinates are the same in every solution (up to elementwise indeterminacies that still do not mix them), then their meaning gets fixed, enabling us to interpret them and intervene on them. Conversely, if we find the latent coordinates not to be identifiable, the theory guides us to what to add in terms of more diversity of data, or additional modelling constraints.

### 2.3. A Unified Causal Lens Predicts Interpretability Failure Modes

Our framework provides language to characterise the relationship between interpretability claims and their supporting evidence. The key question is the gap, if any, between the *asked-for* estimand (what the narrative implies) and the *identified* estimand (what the method actually recovers):

---

[3]The challenge of interpreting an unknown system when no shared reference frame exists to verify meaning, analogous to cryptography without a Rosetta Stone, or encountering an alien civilisation

[4]For instance, consider the reconstruction objective $||\mathbf{x} - q(r(\mathbf{x}))||_2^2$ where $(q, r)$ form an autoencoder pair. Any invertible transformation $a$ would give the exact same value of the objective: $||\mathbf{x} - (q \circ a)(a^{-1} \circ r(\mathbf{x}))||_2^2$ for a different autoencoder pair $(q \circ a, a^{-1} \circ w)$ thus resulting in latent coordinates $a^{-1} \circ r(\mathbf{x})$ instead of $r(\mathbf{x})$.

ASKED-FOR $\tau^\star$ but IDENTIFIED $\hat{\tau}$. Two factors (often compounding) help structure this analysis:

1. **Rung mismatch**: The evidence lives on a lower rung of Pearl's ladder than the claim requires (e.g. associational evidence for an interventional claim).

2. **Identification gap**: Even at the correct rung, the target is identified up to an equivalence class of models, conditional on datasets assumptions and modelling choices.

To measure how well claim language in interpretability papers tracks the evidential strength of the reported methods (i.e., whether a phrase's common interpretation matches its formal causal meaning), we conducted a pilot study described below.

---

**CALIBRATING CLAIM LANGUAGE TO EVIDENTIAL STRENGTH**

We annotated 50 papers (186 claims; protocol in § G) on interpreting model internals, and assigned each claim a method rung (what the reported procedure establishes) and a claim rung (what the surrounding language asserts). Since most papers do not state a causal estimand explicitly, we operationalise claim rungs by mapping common verbs (e.g. "encodes," "mediates") onto the ladder. These verbs often admit causal readings—though they may also reflect disciplinary convention—so the same sentence can be interpreted as making a stronger claim than the reported evidence supports. We therefore mark a *potential* claim–evidence gap when claim rung > method rung. We provide a practitioners' checklist in § G.6 to help align claims with evidentiary support.

**Result.** Roughly half of claims *can* admit a stronger interpretation than the evidence rung licenses—reflecting the field's nascent terminological infrastructure rather than unreliable findings (exact figures in § G). Common patterns include: activation patching results (L2) described with language that can carry counterfactual readings, such as "encodes" or "THE circuit" (L3); probing findings (L1) reported with verbs whose conventional usage may imply stronger causal commitments than the method supports (L3); and single-distribution findings generalised beyond their empirical scope. Primary-annotator confidence drops monotonically with gap size (mean 4.9 for gap-free claims vs. 4.0 for two-rung gaps), consistent with larger rung distances involving harder judgments—though inter-annotator agreement on confidence is near-chance ($\alpha = 0.11$), so this pattern may reflect model-specific calibration rather than an independently validated signal.

---

We use causality to analyse some widely-used interpretability techniques for potential estimand-evidence gaps in Tab. 2. We also acknowledge prior interpretability work raising similar concerns. Next, we analyse representative examples that translate interpetability workflows into a common causal template: the implicit estimand and the evidence along with additional tests that would connect the two (see § F for more). Our goal is to formalise known caveats to diagnose an estimand–evidence gap and the minimal additional evidence that would resolve such that claims will generalise.

---

**CASE STUDY I: SUFFICIENT $\neq$ NECESSARY.**
Activation patching shows that intervening on component set $S$ changes behaviour; the result is often narrated as "$S$ is *the* mechanism for $B$."
**Evidence (L2).** The experiment establishes that intervening on $S$ is sufficient to, on average, shift the output distribution away from the unmodified model:

$$\text{IDENTIFIED} \quad \mathbb{E}_{\mathbf{x}\sim p}\big[p(\mathbf{y} \mid \mathrm{do}(\mathbf{h}_S := \tilde{\mathbf{h}}_S), \mathbf{x})\big]$$
$$\neq \mathbb{E}_{\mathbf{x}\sim p}\big[p(\mathbf{y} \mid \mathbf{x})\big].$$

**Gap.** *The* mechanism asserts necessity and uniqueness, but the evidence establishes neither. Other pathways $S' \neq S$ may be equally sufficient (Wang et al., 2023; McGrath et al., 2023), and sufficiency (intervention changes output) does not entail necessity (without $S$, output *would have been* different).
**What would resolve it (L3).** Necessity is a counterfactual claim—for a specific input, ablating $S$ *would have* changed the output:

$$\text{ASKED-FOR} \quad \mathbf{y}_{\mathbf{h}_S\leftarrow 0}(\mathbf{x}_0) \neq \mathbf{y}(\mathbf{x}_0).$$

This requires a structural model specifying what is held fixed. Uniqueness requires additionally ruling out alternative pathways within a defined mechanism class.
**Takeaway.** Patching identifies only a sufficient (but not necessary) control handle; necessity requires L3 evidence. Establishing uniqueness (ruling out alternatives) is a distinct requirement, and neither is established by L2 evidence. Testing for both denoising (sufficiency) and noising (necessity), as advocated by Heimersheim & Nanda (2024), is one practical step toward closing this gap—their recommendation has a direct theoretical grounding.

---

**CASE STUDY II: PROXY GAMING**
An SAE yields sparse features with high reconstruction accuracy and strong auto-interpretability scores; the result is reported as discovering disentangled or concept-aligned representations.
**Evidence.** The method learns a representation that satisfies the training objective for encoder $\phi$–decoder

$\psi$ pair:

IDENTIFIED $\arg\min_{\phi,\psi} \mathbb{E}[\|\mathbf{a} - \psi(\phi(\mathbf{a}))\|^2] + \lambda\|\phi(\mathbf{a})\|_1$

The solution would minimise reconstruction error while being sparse (as encouraged by the $\ell_1$ sparsity penalty).
**Gap.** Multiple sparse factorisations can achieve similar objective values. Thus, while sparsity is a useful inductive bias, it does not guarantee identification.

ASKED-FOR $\exists \mathbf{c}$ s.t. $\phi(\mathbf{a}) = \mathbf{PDc}$

where $\mathbf{P}$ and $\mathbf{D}$ are permutation and diagonal matrices such that the desired representation identifies the concepts up to permutation and scaling.
**What would resolve it.** Sufficient identifiability conditions under which the recovered representation is unique up to trivial ambiguities (permutation and scaling). For example, Lachapelle et al. (2022) show that if there is sufficient variability across latent variables' contexts, then the representation is identifiable. In practice, this means verifying that the data exhibits such variability and not just that the objective value is low. Joshi et al. (2025) achieve this with SSAEs in practice by uniformly sampling pairs of contexts.
**Takeaway.** Optimising the reconstruction objective and achieving a low $\ell_1-$norm are necessary but not sufficient for identifiability of a concept. It is known that proxies such as reconstruction error or the $\ell_0-$norm cannot be used for model selection (Locatello et al., 2019).

---

**CASE STUDY III: STEERING A CONCEPT**
A steering vector $\mathbf{v}$ reliably shifts behaviour toward refusal or politeness. Usually, this is formulated as *the model represents honesty* or *has a refusal variable*.
**Evidence.** Adding a scaled direction $\alpha\mathbf{v}$ to the activation controllably shifts the output distribution:

IDENTIFIED $p(\mathbf{y} \mid \text{do}(\mathbf{h} := \mathbf{h} + \alpha\mathbf{v}))$ varies with $\alpha$,

where $\alpha \in \mathbb{R}$ denotes steering strength. This establishes that $\mathbf{v}$ affords behavioural control.
**Gap.** Interpreting controllability as encoding implicitly asserts that $\mathbf{v}$ corresponds to an internal causal variable mediating the model's computation for a specific concept:

ASKED-FOR $\mathbf{v}$ corresponds to a causal variable

mediating the computation of $\mathbf{y}$ for a specific concept.

However, $\mathbf{v}$ may entangle multiple concepts, may not be unique, and may exploit distributional shortcuts

rather than the model's natural computation (Turner et al., 2023; Jorgensen et al., 2023; Wu et al., 2025; Mueller et al., 2025).
**What would resolve it.** Evidence that $\mathbf{v}$ is a stable, reusable causal variable—e.g. transfer across distributions (Todd et al., 2023), specificity to a constrained subspace (Marks et al., 2025a), and ideally a causal abstraction in which interventions on $\mathbf{v}$ compose appropriately with other model computations (Geiger et al., 2021). Concurrent work by Miller et al. (2026) demonstrated that an orthogonality regulariser, inspired by the Independent Causal Mechanisms (ICM) principle (Janzing & Schölkopf, 2010), decreased the interference between features under interventions.
**Takeaway.** Steering identifies a control handle on a particular concept, but claiming that the model represents that concept asserts that it is an internal causal variable. The former is supported by L2 evidence, while the latter requires additional structural assumptions beyond controllability.

---

These patterns recur across the interpretability literature (Wang et al., 2023; McGrath et al., 2023; Heap et al., 2025; Turner et al., 2023; Wu et al., 2025; Mueller et al., 2025). In each case, the gap between evidence and claim follows predictably from the rung of the causal hierarchy at which the method operates. Making this explicit clarifies what additional assumptions or experiments are needed to support stronger claims.

## 3. Alternative Views

We briefly situate our framework relative to other prominent views in current interpretability research, highlighting points of alignment and how our framework can contribute additional insights to these perspectives.

### 3.1. Interpretability Can Be Wholly Pragmatic

**Position.** Nanda et al. (2025) argue for a pragmatic criterion: choose proxy tasks such that success would enable progress on an *ultimate goal*. Behavioural outcomes on well-chosen proxies, not internal metrics, determine progress, directly addressing a known failure mode where unsupervised metrics (reconstruction, sparsity, auto-interp scores) provide little evidence about the structure learned by the model (Locatello et al., 2019).

**Rebuttal.** The criterion implicitly assumes what it seeks to avoid: proxy-task success is informative only insofar as the proxy *identifies* structure that is relevant beyond the proxy distribution. Put differently, a proxy can justify stronger conclusions only when it rules out alternative internal explanations that would also solve the proxy yet fail on the target objective. This is an identifiability requirement: the available evidence (here, proxy performance, interventions,

*Table 2.* **Common estimand-evidence gaps in mechanistic interpretability.** Each row contrasts an IMPLIED CLAIM with its ACTUAL SCOPE, as supported by the reported evidence, citing work that documents the gap or demonstrates ways to address it. Our aim is to offer shared terminology that helps unify these efforts and make progress comparable.

| INFERENTIAL GAP | IMPLIED CLAIM | ACTUAL SCOPE |
|---|---|---|
| EXISTENCE → UNIQUENESS | This is *the* circuit or factor *causing* a behaviour. | A circuit that produces output aligned with the behaviour has been found, but the solution is generically non-unique. Other circuits can implement equivalent input–output behaviour (Wang et al., 2023; McGrath et al., 2023). |
| CORRELATION → CAUSATION | Feature $\mathbf{h}_S^{(l)}$ causally mediates behaviour related to concept $c$. | Without targeted interventions (eg. interchange interventions Geiger et al. 2021 or causal scrubbing Chan et al. 2022), we can only determine that $\mathbf{h}_S^{(l)}$ and $c$ are correlated, but the correlation may reflect confounding rather than a causal pathway. |
| DECODABILITY → MODEL USE | The model represents and *computes with* concept $c$. | A probe can decode $c$ from $\mathbf{h}_S^{(l)}$, which does not imply the model's computations depend on $c$ (Belinkov, 2022; Elazar et al., 2021; Hewitt & Liang, 2019; Ravichander et al., 2021). |
| LOCAL SENSITIVITY → GLOBAL CAUSAL ROLE | The mechanism generalises beyond tested prompts. | Sensitivity to $c$ holds for specific inputs since local attributions can be input dependent and may not generalise to held-out distributions (Adebayo et al., 2018; Bilodeau et al., 2024). |
| SUBSPACE → DIRECTION | Concept $c$ is encoded along a single direction in activation space. | A linear probe or PCA component recovers the single most predictive direction for $c$, but it may be best represented through a subspace. Probes trained on different datasets may hence discover different directions depending on which component is most prevalent, each direction a valid but lossy projection of the same higher-dimensional representation (Pan et al., 2025; Engels et al., 2025). |
| SUFFICIENCY → NECESSITY | This component or mechanism is necessary for observed behaviour. | A pathway under specific interventions may be sufficient to produce the desired behaviour, but sufficiency does not entail necessity since alternative pathways for the same behaviour may exist. (Wang et al., 2023; Heimersheim & Nanda, 2024). |
| LOW LOSS → IDENTIFIABILITY | The canonical circuit or factor has been recovered. | The method identifies a solution only up to an equivalence class. Without additional structural constraints, unsupervised methods cannot pin down a unique factorisation (Locatello et al., 2019; Hyvärinen & Pajunen, 1999). |

and stress tests) must constrain the recovered representations such that recovered features are empirically indistinguishable. Recent evidence exposes the gap: SAE probes fail under distribution shift (Kantamneni et al., 2025), which is not a failure of SAEs per se, but of identification [5]—the training distribution did not sufficiently constrain the representation to fix which features should generalise. Identifiability makes this failure predictable rather than surprising, along with additional sufficient conditions to mitigate it.

### 3.2. Symmetries are Sufficient to Formalise Interpretability

**Position.** Concurrent work by Barbiero et al. (2026) define interpretability through symmetry constraints: i.e., a specific set of transformations under which an explanation preserves its meaning. Concretely, if two internal descriptions that are transformations of each other induce the same model behaviour, then any change in the produced explanation under the transformation is attributable to the interpreter's choice

of representation rather than to model-intrinsic structure. This lens complements identifiability: symmetry constraints prescribe which transformations the explanations should be invariant to, and identifiability characterises the equivalence classes of representations that are empirically indistinguishable given available evidence.

**Rebuttal.** Two tensions remain. First, while the framework specifies desirable invariance principles, it leaves underspecified how violations translate into concrete failure modes of existing interpretability methods; making the path from symmetry violation to empirical breakdown explicit would strengthen its operational value. Second, privileging human-interpretable symmetries risks a streetlight effect: model-relevant structure that does not respect these symmetries may be systematically overlooked, even when it materially affects behaviour (see discussion in § 2.2).

## 4. Call to Action.

The failure modes catalogued in § 2.3 can be addressed by recognizing that identifiable causal representation learning (CRL) and interpretability are mutually beneficial. CRL provides formal tools for reasoning about equivalence

---

[5]This is a reason to improve how SAEs are evaluated and constrained, not to discard them. Identifiability conditions offer one solution

classes, causal levels, and transportability, while interpretability supplies empirical phenomena and safety-relevant evaluation criteria. Below we outline research directions that leverage this complementarity, specifying what each field contributes and what concrete work would result.

### 4.1. Counterfactual Semantics for Safety

Safety verification asks counterfactual questions such as: *would this output have been harmful had we not intervened?* Activation patching provides interventional (L2) evidence, whereas counterfactuals (L3) additionally require specifying which variables are held fixed.

CRL OFFERS: Structural models formalising exogenous variables and counterfactual semantics, clarifying when counterfactual quantities are well-defined under explicit abstraction assumptions, c.f., e.g., Geiger et al. (2021). INTERPRETABILITY OFFERS: Realistic architectures (residual streams, attention) that stress-test whether standard structural assumptions hold.

**Research questions:** While counterfactual guarantees (L3) are the ideal target, they are exceptionally difficult to obtain in practice. However, interpretability provides a useful testbed in practice to operationalise what we can ask:

1. When does activation patching coincide with counterfactual conditioning in transformers?
2. For circuits where it does not (e.g., residual-stream coupling), what minimal exogenous annotations restore counterfactual semantics?
3. Can we construct tasks where L2 and L3 answers provably diverge?

### 4.2. Task-Relative Equivalence Classes

Different tasks may require different strengths of identification. While element-wise identifiability is the strongest guarantee and subsumes the other classes, not every task demands it. Steering, for instance, may benefit from block-wise identifiability when related concepts are distributed across dimensions within a shared subspace.

| Task | ASKED-FOR **Minimal equivalence class** |
|---|---|
| Binary classification | Affine (preserves linear separability) |
| Steering | Blockwise (concept subspace) |
| Knowledge editing | Elementwise (needs individual elements) |

CRL OFFERS: A formal vocabulary for equivalence classes induced by symmetries and data constraints. INTERPRETABILITY OFFERS: Empirical evidence (e.g., steering succeeds where editing fails) revealing which equivalence classes are sufficient in practice.

**Research questions:**

1. What are the weakest equivalence classes empirically sufficient for reliable steering or knowledge editing?
2. Can representation learning objectives be designed to target task-specific equivalence classes directly?

### 4.3. Compositional Control

Compositionality varies across intervention types: model edits do not compose (Hoelscher-Obermaier et al., 2023), while task vectors compose linearly (Ilharco et al., 2022). Establishing whether and when steering vectors compose is essential for claiming they can be used to control variables of interest.

CRL OFFERS: Algebraic structure for intervention families (closure, associativity) clarifying when composition is even well-defined. INTERPRETABILITY OFFERS: Empirical compositional failures that reveal violations of these assumptions.

**Research questions:**

1. Under what conditions do representation-level interventions compose additively?
2. Can compositional failures be predicted from representation geometry (e.g., feature overlap or shared support)?
3. Does stronger identifiability correspond to more reliable compositional control? (e.g., Mueller et al. 2025)

### 4.4. Transportability as a Theory of Edit Generalisation

Failures of model editing (Cohen et al., 2024; Hoelscher-Obermaier et al., 2023) are predictable consequences of unexamined transportability assumptions (Bareinboim et al., 2022).

CRL OFFERS: Necessary and sufficient conditions for when causal effects transfer across distributions. INTERPRETABILITY OFFERS: Systematic edit failures as natural experiments revealing transportability violations.

**Research questions:**

1. Can transportability criteria predict which prompts or distributions an edit should generalise to prior to editing?
2. Can editing objectives be designed to optimise for transportable effects rather than in-distribution success?

## 5. Conclusion

Our position is that mechanistic interpretability and identifiable causal representation learning are mutually beneficial. We propose stating interpretability claims and their estimands in the language of causal inference and identifiability (§ 2) to delimit the scope of a method's conclusions and enable comparisons across studies. Potential misinterpretations then reduce to a rung mismatch, an identification gap, or both (§ 2.3)—distinctions our diagnostic checklist (§ G.6) is designed to identify. We further outline four directions in which CRL and mechanistic interpretability can advance one another (§ 4). Our hope is that grounding interpretability in causal inference sharpens our account of what such methods can and cannot establish, a prerequisite for the globally reliable control that safe deployment demands.

**Acknowledgements** The authors thank Thomas Klein, Abhinav Menon for their valuable feedback on the manuscript. Dhanya Sridhar acknowledges support from NSERC Discovery Grant RGPIN-2023-04869, and a Canada-CIFAR AI Chair. Patrik Reizinger acknowledges his membership in the European Laboratory for Learning and Intelligent Systems (ELLIS) PhD program and thanks the International Max Planck Research School for Intelligent Systems (IMPRS-IS) for its support. This work was supported by the German Federal Ministry of Education and Research (BMBF): Tübingen AI Center, FKZ: 01IS18039A. This work was also supported by a grant from Coefficient Giving to Aaron Mueller. Wieland Brendel acknowledges financial support via an Emmy Noether Grant funded by the German Research Foundation (DFG) under grant no. BR 6382/1-1 and via the Open Philanthropy Foundation funded by the Good Ventures Foundation. Wieland Brendel is a member of the Machine Learning Cluster of Excellence, EXC number 2064/1 – Project number 390727645. This research utilized compute resources at the Tübingen Machine Learning Cloud, DFG FKZ INST 37/1057-1 FUGG. Lastly, we thank the organisers and participants of the Fourth Bellairs Workshop on Causality (McGill University Bellairs Research Institute, 14–21 February 2025) for facilitating connections among collaborators.

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

# Contents

# A. Notation and Glossary

## A.1. Formal Definitions: Affordances, Estimands, and Identifiability

This appendix provides formal definitions for the key concepts introduced intuitively in § 2.

### A.1.1. INTERVENTIONS

Interventions are one important class of affordance. We consider interventions targeting inputs $\mathbf{x}$, activations $\mathbf{a}^{(l)}$, representations $\mathbf{h}^{(l)}$, or model parameters $\theta$; denote the set of admissible targets $v$ by $\mathcal{V} := \{\mathbf{x}\} \cup \{\mathbf{a}^{(l)}\}_{l=1}^{L} \cup \{\mathbf{h}^{(l)}\}_{l=1}^{L} \cup \{\theta\}$ denote the set of admissible intervention targets. For any such $v \in \mathcal{V}$, we write $\mathrm{do}(v = v')$ to denote setting $v$ to $v'$; e.g. clamping, activation patching, zero-ablating. More generally, $\mathrm{do}(\mathcal{I}_v)$ would represent modifications such as steering, scaling, fine-tuning, which don't fix the target to a specific value. This notation unifies input-, representation-, activation-, and parameter-level manipulations within a single calculus.

*Table 3.* Examples of intervention classes in LLMs and representative methods.

| INTERVENABLE OBJECT | REPRESENTATIVE METHODS |
|---|---|
| **INPUTS**
*prompts, context* | Minimal pairs (Marvin & Linzen, 2018) · Prompt perturbations (Ribeiro et al., 2020) · Jailbreak suffixes (Zou et al., 2023) · Persona modulation (Zheng et al., 2024) |
| **ACTIVATIONS**
*layer states* | Activation patching (Wang et al., 2023) · Causal mediation analysis (Vig et al., 2020) · Ablations (Ghorbani & Zou, 2020) |
| **REPRESENTATIONS**
*learned features* | SAE feature steering (Templeton et al., 2024)· Interchange interventions (Geiger et al., 2022) |
| **PARAMETERS**
*weights* | ROME (Meng et al., 2022)· MEMIT (Meng et al., 2023)· ReFT–r1 (Wu et al., 2025) · LoRA (Hu et al., 2022) |

### A.1.2. COUNTERFACTUALS

Counterfactual queries ask how the output *would have* differed under an alternative intervention, given what was actually observed. For an observed input–output pair $(\mathbf{x}, \mathbf{y})$ and intervention target $v \in \mathcal{V}$, we write $\mathbf{y}_{v \leftarrow v'}$ (or, $\mathbf{y}_{\mathcal{I}_v}$) to denote the counterfactual output that would have been produced had the intervention $\mathrm{do}(v = v')$ (or, induced by $\mathcal{I}_v$) been applied, while holding all other external factors fixed.

## A.2. Features

This section consolidates the various meanings of "feature" used in mechanistic interpretability and causal representation learning, clarifying terminological ambiguities that can lead to confusion when bridging these communities.

### A.2.1. TWO NOTIONS OF FEATURE

Contemporary research on neural network interpretability operates with distinct notions of what constitutes a *feature*. Mechanistic interpretability typically considers a feature as any semantically meaningful, extractable piece of information encoded in model activations (Elhage et al., 2022; Bricken et al., 2023). This operational definition carries no strong ontological commitment regarding the existence of latent generative factors or an underlying data-generating process (DGP).

### A.2.2. FEATURE OPERATIONALISATIONS IN MECHANISTIC INTERPRETABILITY

In mechanistic interpretability, a *feature* may refer to several different objects, with the common theme being they are all internal intervenable units of the model. This terminological ambiguity reflects genuine disagreement about the right level of analysis for understanding neural networks:

1. **Individual neurons**: Early work studied when "individual neurons correspond to natural 'features' in the input" (Jermyn et al., 2022), treating single units $a_i^{(l)}$ as the atomic interpretable components. However, the prevalence of polysemantic neurons—where single neurons respond to multiple unrelated concepts—has challenged this view (Elhage et al., 2022).

2. **Linear directions**: The linear representation hypothesis formalises features as directions $\mathbf{v}$ in activation space such

that $\mathbf{v}^T\mathbf{a}$ varies systematically with some property of interest (Park et al., 2024). This view underlies concept activation vectors (Kim et al., 2018) and activation steering (Turner et al., 2023).

3. **Dictionary elements**: The dominant current usage refers to basis directions in the learned encoding space of a sparse autoencoder (SAE), motivated by the superposition hypothesis—that models represent more features than they have dimensions (Elhage et al., 2022; Bricken et al., 2023; Cunningham et al., 2023).

4. **Circuit components**: In circuit-based analysis, features may refer to the nodes in sparse feature circuits—causally implicated subnetworks that explain model behaviours (Marks et al., 2025a).

These different operationalisations carry different assumptions about what makes a unit interpretable.

In contrast, causal representation learning (CRL) presupposes the existence of ground-truth latent factors $\mathbf{z} \in \mathbb{R}^{d_z}$ that causally generate observations $\mathbf{x} \in \mathbb{R}^{d_x}$ through some mixing function $g : \mathbb{R}^{d_z} \to \mathbb{R}^{d_x}$ (Schölkopf et al., 2021; Locatello et al., 2019). The central question in CRL is whether these latent factors can be *identified*—that is, recovered up to well-characterised equivalence classes—from observations alone or with auxiliary information such as temporal structure (Hyvärinen & Morioka, 2016; 2017), paired samples (Locatello et al., 2020), or multi-environment data (Ahuja et al., 2023).

### A.3. Three Sources of Identifying Structure

The strategies outlined in the main text—interventions, natural variation, and inductive biases—correspond to three recurring sources of identifying structure in the representation learning literature:

**Interventional data.** Paired samples before and after perturbations, temporally resolved actions, or multiple environments with unknown targets can force competing explanations apart, identifying latent variables up to equivalence classes determined by the intervention family (Eberhardt & Scheines, 2007; Pearl, 2009; Brehmer et al., 2022; Lippe et al., 2022; von Kügelgen, 2024; Varıcı et al., 2024).

**Natural variation.** Natural variation provides similar leverage without explicit intervention: nonstationarity enables nonlinear ICA through segment discrimination (Hyvärinen & Morioka, 2016; 2017; Khemakhem et al., 2020a), while self-supervision and weak supervision exploit contrasts across timesteps, augmentations, or contexts (von Kügelgen et al., 2021a; Ahuja et al., 2022).

**Inductive biases.** Inductive biases further constrain the equivalence class by ruling out families of solutions: independence assumptions (Comon, 1994; Hyvärinen et al., 2001; Gresele et al., 2021), sparsity constraints (Lachapelle et al., 2022), and functional restrictions on the mixing process (Khemakhem et al., 2020b; Roeder et al., 2021).

#### A.3.1. UNIFYING PERSPECTIVES

The Identifiable Exchangeable Mechanisms (IEM) framework (Reizinger et al., 2025) provides a unifying perspective on these sources of variation, distinguishing two complementary forms: *cause variability*, where the distribution of inputs or latent sources varies across contexts while the mechanism mapping them to outputs remains fixed, and *mechanism variability*, where inputs are held constant while the generative mechanism changes.

In practice, identifiability results typically combine these sources—e.g. temporal structure plus sparsity (Hyvärinen & Morioka, 2017; Lippe et al., 2022), weak supervision plus independence (Locatello et al., 2020; Ahuja et al., 2022), or interventions plus mechanism constraints (Brehmer et al., 2022; Lachapelle et al., 2022)—and the resulting equivalence class reflects the joint constraints imposed by all available affordances.

## B. Philosophical Commitments

### B.1. The Metaphysics of Affordance-based Identifiability

A common (often implicit) premise in identifiable (causal) representation learning is that there exists a *privileged* latent description of the world, an "ideal reality" given by a true set of generative factors and that a learned representation should recover it, up to a narrow equivalence class (typically permutation and rescaling). Such a representation is called *platonic*, harking back to Plato's theory of "forms" (Plato, -380; Kraut, 2022)—eternal and changeless entities paradigmatic of the nature of the perceived world.

Structuralist traditions motivate a different stance. Rather than treating latent variables as privileged ontological objects whose coordinates we only imperfectly recover, structuralism holds that scientific knowledge concerns *relations, invariants, and transformations*, not things-in-themselves. On this view, the epistemic content of a theory lies in the structure it preserves across admissible representations, while the nature of the underlying entities remains either inaccessible or underdetermined.

This position has appeared in multiple guises. In the philosophy of science, Poincaré (1905) argues that while ontological posits may change across scientific revolutions, certain *relations* (expressed through equations, symmetries, and invariants) persist; this privileges epistemology over ontology, and suggests that stable knowledge resides in structure rather than in a unique set of underlying entities. In mathematics, the Bourbaki programme (Bourbaki, 1950) formalised structuralism by defining objects only up to isomorphism: groups, spaces, and algebras are not collections of elements with intrinsic identity. In philosophy, Kant's critical realism (Kant, 1781) similarly denies access to noumena, or the world as it is., while affirming that stable structure can nevertheless be known through forms of interaction.

Read through this lens, one can argue that the practice of identifiability already aligns more closely with structural realism than with a strong Platonic metaphysics. Identifiability results do not recover a unique latent state of the world; they recover an equivalence class under a transformation group that preserves specified relations. Permutation, rescaling, and invertible reparameterisations are not technical pathologies but explicit acknowledgements that only certain aspects of a representation are empirically meaningful. The identifiable object is not a latent variable *per se*, but the structure defined by its invariants.

The apparent Platonic commitment enters only through interpretation. Latent variable models are often narrated as approximations to a true generative process, giving rise to the impression that identifiability concerns proximity to an underlying set of hidden entities. A structuralist reading weakens this ontological claim without weakening the theory. The equivalence class need not correspond to "the true ground-truth generative factors"; it need only faithfully represent the relational structure that is recoverable from data and stable across environments. In this sense, identifiability is best understood as a statement about *what structure is invariantly identifiable, not about what entities exist.*

Affordances (Gibson, 1979) provide a concrete operationalization of this structuralist view. In Gibson's ecological psychology, an affordance is not a property of an object in isolation but a relation jointly defined by an agent and its environment: what actions the pairing makes possible. What an object *is*, functionally speaking, is inseparable from what it *affords*. Translating this to representation learning: a latent dimension is meaningful insofar as it parameterises stable affordances—predictable, reproducible changes in observations under manipulation. Identifiability, then, concerns the recovery of coordinates that preserve these affordances, not the recovery of hidden objects independent of any interaction.

Under this view, the goal of identifiable representation learning shifts. Rather than asking how closely a learned representation approximates presumed ground-truth factors, we ask whether it preserves the same affordance structure under admissible transformations. The key insight is threefold: (i) affordances define which variations are observable or inducible; (ii) these variations break equivalence classes among candidate representations such that two representations that behave identically under all afforded interactions are, for our purposes, equivalent;(iii) identifiability characterises precisely which structure survives this winnowing. Hence a representation is structurally sufficient if it supports all affordance-based distinctions the analyst can induce, even without recovering any unique "true" latent variables.

Identifiability thus becomes a theory of structural sufficiency—it characterises the maximal structure that can be stably recovered while remaining agnostic about deeper ontological commitments. This reframing dissolves a persistent confusion. Latent variables are not the necessary *content* of ICA or related models; they are convenient *coordinates* for representing the recovered structure. Just as Cartesian and polar coordinates describe the same geometric relations, different latent parameterizations may describe the same affordance structure. This addresses a central question: what does identifiability mean when privileged ground truth is never accessible? It specifies which structural distinctions are recoverable from the interactions available. From this perspective, the affordance set is induced by the experimental conditions under which data are generated: auxiliary variables, multiple environments, temporal structure, or admissible interventions. We need not presuppose a ground-truth generative process; the affordances themselves define what can be learned.

This motivates *affordance-based identifiability*: assess a representation not by axis-aligned recovery of latent coordinates, but by the invariances and interventional queries it supports.

### B.2. Pragmatism

**Core idea.** Pragmatism is a philosophical tradition—developed by James (1907) and Dewey (1948)—that evaluates ideas primarily by their practical consequences rather than by correspondence to abstract truth. On this view, the meaning of a concept is inseparable from the difference it makes in practice: if two theories yield identical predictions and interventions, they are pragmatically equivalent, regardless of their metaphysical commitments.

**ML context.** When interpretability researchers ask "does this explanation help us predict model failures?" or "can we use this feature to steer behavior?", they are implicitly adopting a pragmatist stance. A circuit explanation that enables reliable intervention is pragmatically valuable even if we cannot prove it reflects the model's "true" computation. The challenge is that pragmatism without discipline can become an excuse for vagueness—claiming success on easy proxies while avoiding harder questions about generalization.

### B.3. Constructive Empiricism

**Core idea.** Constructive empiricism, articulated by Bas (1980), holds that science aims to produce theories that are *empirically adequate*—that "save the phenomena" by correctly predicting observable outcomes—without necessarily claiming to describe unobservable reality as it truly is. We can accept a theory as useful without believing it literally describes hidden mechanisms.

**ML context.** Consider a sparse autoencoder (SAE) that decomposes activations into interpretable features. A constructive empiricist would say: we can use these features for prediction and control without claiming they represent the model's "real" internal concepts. The SAE is a useful instrument for organizing our observations, not necessarily a window into ground truth. This perspective is liberating (we need not solve the "what are concepts really?" question) but also demands honesty about what our tools actually deliver.

### B.4. Underdetermination

**Core idea.** Underdetermination refers to situations where multiple, mutually incompatible explanations are equally consistent with all available evidence (Quine, 1975). No amount of data can uniquely determine which explanation is correct—additional assumptions or constraints are required to break the tie.

**ML context.** This problem pervades interpretability. A linear probe achieving 95% accuracy on sentiment classification does not uniquely identify "the sentiment direction"—many directions may achieve similar performance, and the probe may exploit correlates rather than causes. Similarly, multiple circuit hypotheses may explain the same input-output behavior. Underdetermination is not a failure of method but a structural feature of inference from finite evidence. The remedy is not to ignore it but to explicitly characterise the equivalence class of solutions and understand what additional evidence (e.g. interventions, distribution shifts) could narrow it.

### B.5. Ontological Faithfulness vs. Instrumental Adequacy

**Core idea.** These terms distinguish two standards for evaluating explanations. *Ontological faithfulness* asks whether an explanation accurately describes what is "really there"—the true structure, mechanisms, or entities underlying a phenomenon. *Instrumental adequacy* asks only whether the explanation serves its intended purpose—enabling prediction, control, or communication—without commitment to ontological accuracy (Potochnik, 2017).

**ML context.** When we identify a "deception feature" in an LLM, are we discovering something the model genuinely represents, or constructing a useful fiction that helps us predict and steer behavior? Ontological faithfulness would require the feature to correspond to some real computational primitive; instrumental adequacy requires only that interventions on this feature reliably produce the desired effects. Much interpretability research implicitly claims ontological faithfulness ("the model *has* this concept") while only demonstrating instrumental adequacy ("this probe *predicts* this label"). Being explicit about which standard we are meeting helps calibrate the strength of our claims.

**Literature evidence.** The gap between these standards is well-documented. Hewitt & Liang (2019) introduced "control tasks"—random label assignments that probes can nonetheless fit—demonstrating that high probe accuracy does not entail that representations *encode* the probed property; the probe may simply memorise. Elazar et al. (2021) showed that probing accuracy is *not correlated* with task importance: properties easily extracted by probes may not be *used* by the model, calling for "increased scrutiny of claims that draw behavioral or causal conclusions from probing results." More recent work on sparse autoencoders (SAEs) reveals analogous concerns: Engels et al. (2024) find substantial "dark matter"—unexplained variance that SAE features fail to capture—questioning whether SAE features constitute the model's "true" computational primitives. Chanin et al. (2025) show that SAEs trained on LLMs suffer from "feature hedging," merging correlated features in ways that destroy the monosemanticity they are meant to provide.

**Example.** Consider the common claim that "BERT encodes part-of-speech information." A linear probe trained on BERT's hidden states achieves >97% accuracy at predicting POS tags (Hewitt & Manning, 2019). This is often interpreted ontologically—as if BERT internally represents parts-of-speech as a computational primitive. However, Hewitt & Liang (2019) showed that probes with similar capacity can achieve high accuracy on *random control labels* that carry no linguistic content, revealing substantial probe memorization. More strikingly, Elazar et al. (2021) demonstrated that removing the information exploited by POS probes via causal intervention ("amnesic probing") had minimal effect on BERT's language modeling performance—suggesting that while the information is *extractable* (instrumental adequacy), it may not be *used* by the model's actual computation (calling into question ontological faithfulness). The probe succeeds, but claiming BERT "has" POS representations in any strong sense overstates what the evidence supports.

### B.6. Transportability

**Core idea.** Transportability, formalised in causal inference by Pearl & Bareinboim (2014) and Bareinboim & Pearl (2012), concerns when causal relationships learned in one setting (population, environment, distribution) remain valid in another. A causal effect is transportable if we can formally justify applying it to a new context, accounting for differences between source and target.

**ML context.** Suppose we discover a steering vector that induces honesty in GPT-4 on a particular distribution of prompts. Does this vector work on different prompt types? On other models? In deployment conditions not seen during development? Transportability formalises these questions. A steering intervention that only works in-distribution has limited practical value; one that transports across contexts provides robust control. The causal inference literature provides tools for reasoning about when and why transport succeeds or fails—tools that interpretability research has largely not yet adopted.

## C. Technical Review

### C.1. Background: Identifiability Results from Representation Learning

This section provides technical background on identifiability results. These results formalise how different sources of variation—interventions, natural distribution shifts, and inductive biases—constrain the equivalence class of recoverable latent structures.

#### C.1.1. REINTERPRETING CLASSICAL IDENTIFIABILITY

Classically, identifiability results are stated with respect to latent "ground-truth" variables that generate observed data. We adopt a different reading: ground-truth variables are not the ontologically privileged entities that exist independently in the observations. Rather, they are formal reference variables used to specify which distinctions are, in principle, recoverable from a given interface—i.e. from the affordances (variations and assumptions) available to an interpreter. On this reading, identifiability characterises an *equivalence class* of internal structures (Gresele et al., 2021; Lachapelle et al., 2022; Ahuja et al., 2022; Guo et al., 2024; Reizinger et al., 2025) consistent with those affordances, not a unique ontology of concepts.

### C.2. Background: Reconciling Mechanistic Interpretability and Causal Representation Learning

The term "feature" carries different connotations in mechanistic interpretability and causal representation learning; see § A.2 for a detailed discussion of these two notions and how they can be reconciled.

#### C.2.1. THE DGP AS A SPECIFICATION OF INTERESTINGNESS

One reconciliation of these perspectives, following Lachapelle et al. (2024), is to view the DGP assumption not as a metaphysical claim about platonic reality, but as a *mathematical specification of what makes features interesting*—a position that aligns with the structuralist reading of identifiability developed in § B.1. Different choices of DGP correspond to different criteria for feature quality:

- **Independence:** Classical ICA (Hyvärinen et al., 2001) defines interesting directions as those maximizing statistical independence of the recovered components.
- **Variation across contexts:** Paired-sample and multi-view approaches (Locatello et al., 2020; von Kügelgen et al., 2021b) identify features that vary across paired observations while others remain fixed.
- **Intervention-sensitivity:** Causal approaches (Ahuja et al., 2023; Buchholz et al., 2024) privilege factors whose distributions shift under interventions or across environments.

Under this view, identifiability theory provides mathematical precision to the otherwise vague notion of "semantically meaningful features" commonly invoked in mechanistic interpretability.

#### C.2.2. ICA WITHOUT LATENT VARIABLES: A PHILOSOPHICAL NOTE

An important observation, due to Hyvärinen et al. (2001), is that ICA admits formulations that do not posit latent variables at all. One can define the ICA problem purely in terms of finding "maximally independent" projections of observed data, without reference to an underlying generative process. In the linear case, the equivalence between the generative and projection-based formulations is well-established; in the nonlinear case, analogous results hold under suitable definitions of independence (Khemakhem et al., 2020a, Appendix F).

This distinction parallels a broader divide between latent variable models—which assume unobserved generative factors—and energy-based models, which make no such commitments and are, in a sense, "assumption-free" (LeCun et al., 2006). However, the latent variable formulation naturally admits a *causal* interpretation: the standard ICA equation $\mathbf{x} = A\mathbf{s}$ should

be read as an assignment $\mathbf{x} := A\mathbf{s}$ (in the sense of structural causal models), indicating that latent sources *causally generate* observations rather than merely being correlated with them (Pearl, 2009).

### C.2.3. IMPLICATIONS FOR MECHANISTIC INTERPRETABILITY

Grounding interpretability methods in identifiability theory offers several potential benefits:

**Consistency across methods.** Even without commitment to a "true" DGP, identifiability results can address whether different interpretability methods—e.g. sparse autoencoders with varying architectures (Cunningham et al., 2023), linear probes (Alain & Bengio, 2017), or distributed alignment search (Geiger et al., 2024)—converge on equivalent feature representations. This question of *inter-method consistency* is distinct from, and arguably more tractable than, the question of whether any method recovers ground-truth factors (Roeder et al., 2021).

**Operationalising *semantic meaning*.** Mechanistic interpretability literature frequently appeals to features being "semantically meaningful" or "human-interpretable" without formal criteria for these properties. Identifiability theory can provide such criteria—statistical independence, intervention-sensitivity, or invariance across domains—that operationalise interestingness in falsifiable terms.

**Downstream guarantees.** A causal DGP assumption may yield guarantees that purely statistical notions cannot provide. Recent empirical work has documented failures of sparse autoencoder features to transfer across distribution shifts (Makelov et al., 2024) or to support reliable steering interventions (DeepMind Safety Research, 2024), suggesting that features identified without causal grounding may lack the robustness required for safety-critical applications.

### C.2.4. ABSTRACTION AND SELECTION IN LEARNED REPRESENTATIONS

Both the specification of a DGP and the learning process itself can be understood as performing two complementary operations (Rubenstein et al., 2017; Beckers & Halpern, 2019):

1. **Abstraction:** Selecting the level of detail at which to represent the world (e.g. modeling objects as point masses versus textured 3D entities).
2. **Selection:** Discarding factors deemed irrelevant to the task at hand (e.g. ignoring friction in a mechanics problem).

Under this framing, mechanistic interpretability can be viewed as the problem of reconstructing the *learned* DGP implicit in a trained model's representations. Crucially, this learned DGP may differ from any "true" external DGP due to inductive biases, architectural constraints, or training dynamics—a form of model misspecification that interpretability methods must contend with.

### C.3. The causal ladder for interpretability queries

Pearl's causal ladder (Pearl, 2009) distinguishes three rungs of evidence: association (L1), intervention (L2), and counterfactual (L3). This hierarchy provides a principled framework for classifying what interpretability methods can actually establish. In practice, interventions such as activation patching or steering do not simply "change a variable", but also implicitly change how the rest of the network processes it. Consequently, the observed effect may reflect properties of the intervention procedure itself (e.g. where the patch is taken from, how it is injected, or how strongly it is applied), rather than the effect of the variable in isolation.

§ D provides a comprehensive reference that maps interpretability queries to their ladder rungs (Tab. 4), with detailed examples illustrating how to locate your analysis on the hierarchy.

## D. The Causal Ladder: A Unified Reference

This section provides a comprehensive reference for Pearl's causal ladder (Pearl, 2009) as applied to mechanistic interpretability. For technical background on identifiability results and the reconciliation of mechanistic interpretability with causal representation learning, see § C.1. We first present a systematic mapping of interpretability queries to their ladder rungs (Tab. 4), then provide detailed guidance for locating your own analyses on this hierarchy.

### D.1. Interpretability Queries Across the Ladder

Tab. 4 maps common interpretability questions onto the three rungs of Pearl's ladder. Each rung corresponds to a distinct type of evidence: association (L1), intervention (L2), and counterfactual (L3). The table organises queries by what is being examined—inputs, activations, representations, or parameters—and illustrates representative methods at each level.

*Table 4.* Interpretability queries across Pearl's causal ladder, expressed in terms of association, intervention, and counterfactuals.

| Rung of the Ladder | Query Distribution | Practical Query (with Examples) |
|---|---|---|
| **1: Association** | $p(\mathbf{y} \mid \mathbf{x})$ | *Prediction:* Given an input prompt $\mathbf{x}$, what output tokens $\mathbf{y}$ are likely? (Standard next-token prediction, logit inspection) |
| | $p(\mathbf{h} \mid \mathbf{x})$ | *Encoding:* Given an input $\mathbf{x}$, what internal representations or features $\mathbf{h}$ are typically activated? (Activation logging, SAE feature attribution, probing) |
| | $p(\mathbf{y} \mid \mathbf{h})$ | *Decoding:* Given an observed internal trace $\mathbf{h}$, what outputs $\mathbf{y}$ are likely? (Linear probes, sparse decoding, feature-to-token analyses) |
| **2: Intervention** | $p(\mathbf{y} \mid \mathrm{do}(\mathbf{x} = \mathbf{x}'))$ | *Input intervention:* If I change the prompt from $\mathbf{x}$ to $\mathbf{x}'$, how does the output change? (Minimal pairs, prompt perturbations, jailbreak tests) |
| | $p(\mathbf{y} \mid \mathrm{do}(\mathbf{a}^{(l)} = \mathbf{a}'))$ | *Activation intervention:* If I overwrite or patch activations at layer $l$, how does the output change? (Activation patching, causal tracing, ablations) |
| | $p(\mathbf{y} \mid \mathrm{do}(\mathbf{h} = \mathbf{h}'))$ | *Representation intervention:* If I manipulate a feature or direction in representation space, what behavior changes? (SAE feature steering/ablation, DAS interchange interventions) |
| | $p(\mathbf{y} \mid \mathrm{do}(\theta = \theta'))$ | *Parameter intervention:* If I edit the model's weights, does the model's behavior change as intended? (ROME, MEMIT) |
| **3: Counterfactual** | $\mathbf{y}_{\mathbf{x} \leftarrow \mathbf{x}'}$ | *Input counterfactual:* Given this specific run, would the model's output have differed had the prompt been $\mathbf{x}'$ instead of $\mathbf{x}$? (Prompt counterfactual analysis) |
| | $\mathbf{y}_{\mathbf{a}^{(l)} \leftarrow \mathbf{a}'}$ | *Activation counterfactual:* For this exact generation, would the output have changed if activations at layer $l$ had been different? (causal scrubbing-style analyses) |
| | $\mathbf{y}_{\mathbf{h} \leftarrow \mathbf{h}'}$ | *Representation counterfactual:* Would this behavior still have occurred if a specific internal feature had been absent or altered? (Feature-level necessity tests) |
| | $\mathbf{y}_{\theta \leftarrow \theta'}$ | *Model counterfactual:* Would this same input have produced a different output if the model's parameters had encoded different knowledge? |

## D.2. Locating an Interpretability Method on the Causal Ladder

The following diagnostic questions help practitioners identify what rung their evidence actually occupies. Each rung is illustrated with a concrete example, specifying what the evidence licenses and what it does not.

**L1: Association—Is the system manipulated, or only observed?** If the method only computes statistics from forward passes—e.g. probing, PCA, feature attribution—the evidence is associational (L1) evidence, even if the summary is interpreted causally.

*Example: Linear probing for sentiment.* Suppose we train a linear classifier on layer-12 activations to predict sentiment labels. The probe achieves 92% accuracy on held-out data.

 (i) *Setup:* Collect activations $\{\mathbf{a}_i^{(12)}\}$ from forward passes on sentiment-labeled text.
 (ii) *Method:* Fit logistic regression $\hat{y} = \sigma(\mathbf{w}^\top \mathbf{a}^{(12)} + b)$ to predict positive/negative.
(iii) *Result:* High accuracy shows sentiment information is *decodable*—an external classifier can extract it.

*What L1 licenses:* "Sentiment is linearly decodable from layer 12." *What L1 does not license:* "The model represents sentiment at layer 12" or "Layer 12 computes sentiment." Decodability $\neq$ encoding; the information may be present but unused by the model's own computation.

**L2: Intervention—Is the manipulation externally controlled?** If the method modifies the forward pass—clamping an activation, patching from a different input, ablating a component, adding a steering vector—and measures the downstream effect, the evidence is interventional (L2). The key is that the modification is set by the experimenter, not determined by the natural computation.

*Example: Activation patching for indirect object identification.* We investigate whether a specific attention head mediates the IOI task (completing "Mary gave a book to..." with the indirect object).

 (i) *Setup:* Run the model on prompt $A$ ("Mary gave a book to John. Mary gave a pencil to...") and record activations. Run on corrupted prompt $B$ where names are swapped.
 (ii) *Intervention:* At head 9.6, replace the activation from run $A$ with the activation from run $B$: $\mathbf{a}_A^{(9.6)} \leftarrow \mathbf{a}_B^{(9.6)}$.
(iii) *Measurement:* Observe whether the output changes from "John" toward "Mary."
 (iv) *Result:* If patching head 9.6 substantially changes the output, we have evidence that this head *causally contributes* to the IOI behavior under this intervention.

*What L2 licenses:* "Intervening on head 9.6 changes IOI behavior under these tested conditions." *What L2 does not license:* "Head 9.6 is the IOI mechanism" (sufficiency $\neq$ necessity; other paths may exist) or "This finding will hold on all IOI-like prompts" (requires transportability testing).

**L3: Counterfactual—Is the query anchored to a specific observation?** Counterfactual claims reason about the *same* instance under an alternative intervention: "for this input that produced $\mathbf{y}$, what would the output have been had $\mathbf{a}^{(l)}$ taken value $\mathbf{a}'$?" This demands not just an intervention, but a model of how latent variables would have responded—requiring structural assumptions beyond what patching alone provides.

*Example: The Eiffel Tower counterfactual.* An LLM processes "The Eiffel Tower is in" and outputs "Paris." We observe activations $\{\mathbf{a}^{(l)}\}$ throughout. The counterfactual question is: *what would the model have output if $\mathbf{a}^{(5)}$ had taken the value it takes when processing "The Colosseum is in"?*

Operationally, counterfactual reasoning follows the *abduction–action–prediction* recipe (Pearl, 2009):

(i) *Abduction:* Condition on the observed run—input, output, and the full trace $\{\mathbf{a}^{(l)}\}$—to infer what latent or exogenous factors explain *this particular* forward pass. In our example: what about the model's internal state, beyond the activations we recorded, determined that "The Eiffel Tower is in" $\rightarrow$ "Paris"?

(ii) *Action:* Intervene on the target variable. Here, set $\mathbf{a}^{(5)} \leftarrow \mathbf{a}'^{(5)}$, the activation from the Colosseum prompt.

(iii) *Prediction:* Propagate forward under the intervention, *holding fixed* the exogenous factors inferred in step (i), to compute the counterfactual output $\mathbf{y}_{\mathbf{a}^{(5)} \leftarrow \mathbf{a}'^{(5)}}$.

*What L3 licenses:* "For this specific forward pass, had $\mathbf{a}^{(5)}$ taken the Colosseum value, the model would have output 'Rome.'"

*What L3 does not license:* Generalisations to other inputs (requires separate analysis or transportability assumptions) or mechanism-level claims like "layer 5 encodes location." Crucially, L3 requires a structural causal model specifying how latent variables interact—not merely the ability to intervene. Standard activation patching performs step (ii) but lacks the structural model needed for steps (i) and (iii). This is why most "counterfactual" claims in interpretability are actually L2 claims narrated counterfactually.

Tab. 1 maps the method families from § E onto this hierarchy, contrasting the question each method is typically recruited to answer with the question its evidence can actually support. A recurring pattern emerges: interpretability claims often climb the ladder without the requisite evidence. Most interpretability methods that manipulate internal states license L2 evidence (Vig et al., 2020; Elhage et al., 2021; Chan et al., 2022; Turner et al., 2024). Achieving L3 generally requires stronger modeling assumptions about latent variables and mechanisms (Geiger et al., 2021). As a result, claims framed counterfactually—e.g. "this feature caused the output" or "the model would not have produced $y$ without this component"—often rest on L1 or L2 evidence, thereby inflating the rungs.

## E. Implicit Estimands in Interpretability Methods

We examine five method families, identifying for each: the question the method is typically recruited to answer, the quantity it actually estimates, and the gap between the two. We have also used identifiability as a diagnostic lens, asking whether a method's evidence pins down its claimed quantity. Tab. 5 provides a summary; we elaborate below.

**Feature Attribution.** Saliency maps (Simonyan et al., 2014), Gradient×Input (Shrikumar et al., 2017), Integrated Gradients (Sundararajan et al., 2017), SHAP (Lundberg & Lee, 2017), and LIME (Ribeiro et al., 2016) are framed as answering a counterfactual question: *which input features are responsible for this output?* (Bilodeau et al., 2024). In practice, these methods estimate local sensitivity—the rate of change of output under input perturbations—via gradients (Simonyan et al., 2014; Ancona et al., 2018) or surrogate fits (Ribeiro et al., 2016). SHAP targets Shapley values over feature coalitions, a distinct quantity requiring marginalization under analyst-defined baselines. The evidence is associational throughout: no intervention on the data-generating process occurs. Failure modes follow predictably: saliency maps remain stable under parameter randomization (Adebayo et al., 2018), attributions shift under semantics-preserving transformations (Kindermans et al., 2019), and SHAP values distribute arbitrarily among correlated features (Merrick & Taly, 2020).

**Probing and Diagnostic Methods.** Linear probes (Hewitt & Manning, 2019; Alain & Bengio, 2017), concept activation vectors (TCAVs; Kim et al., 2018), and the logit lens (nostalgebraist, 2020) ask whether a concept is *encoded* in the model's representations. What they estimate is narrower: the existence of a linear decision boundary separating concept–positive from concept–negative examples in activation space. Thus, two models are indistinguishable if they admit the same linear

*Table 5.* Interpretability method families and the implicit estimands they target.

| METHOD FAMILY | REPRESENTATIVE METHODS | IMPLICIT ESTIMAND | EVIDENCE TYPE |
|---|---|---|---|
| **FEATURE ATTRIBUTION** *saliency / scores* | Saliency · Grad×Input · Integrated Gradients · SHAP · LIME | Local input–output sensitivity (at a given input and parameterization); marginal contribution relative to an analyst-defined baseline or perturbation distribution | Gradients / synthetic perturbations |
| **DIAGNOSTICS / PROBING** *readouts* | Linear probes · CAVs · Logit lens | Existence of a decision function in a restricted probe class that predicts a concept from internal states; correlational alignment with external labels | Supervised labels / correlations |
| **CAUSAL / CIRCUIT-BASED** *interventions* | Activation patching · Causal tracing · Circuit analysis | Interventional effect of manipulating internal states (e.g. replacing $\mathbf{a}^{(l)}$) on downstream outputs | Controlled interventions |
| **STRUCTURED DECOMPOSITION** *basis learning* | SAEs · PCA · Dictionary learning | Factors induced by an optimisation objective (e.g. reconstruction with sparsity or low-rank constraints) | Reconstruction objective |
| **THEORY-GROUNDED** *identifiable structure* | DAS · SSAEs | Recovery up to an equivalence class under stated identifiability assumptions | Interventions / Sufficient variations / Inductive biases |

separator, regardless of whether either model actually uses the concept downstream. The equivalence class thus contains both models that genuinely encode the concept (it plays a causal role in computation) and models where the concept is merely *recoverable* as a correlate of structure the model uses for other purposes. The claimed quantity—whether the concept is represented—is not identified; only decodability is. Breaking this equivalence class requires interventional evidence: if we intervene on the concept representation and observe its downstream effects, we move from asking "can we decode this concept?" to "does the model use it?".

**Causal and Circuit-Based Methods.** Activation patching (Elhage et al., 2021), causal tracing (Meng et al., 2022), and circuit analysis (Olah et al., 2020; Conmy et al., 2023) aim to identify which components *uniquely mediate* behavior. The estimand—the causal effect of replacing an activation under $\mathrm{do}(\mathbf{a}^{(l)} = \mathbf{a}'^{(l)})$—is genuinely interventional (Vig et al., 2020; Pearl, 2001). The affordance family $\mathcal{U}$ now includes controlled interventions. However, multiple circuits can produce identical patching effects: the observation that patching component $C$ restores behavior does not rule out alternative components $C'$ that would do the same. Under $\mathcal{U}$, models with different "true" mediating structure are indistinguishable if they respond identically to the interventions performed. Existence of a sufficient path is identified; uniqueness and necessity are not (Zhang & Nanda, 2024; Chan et al., 2022).

**Structured Decomposition.** Sparse autoencoders (SAEs; Cunningham et al., 2023; Bricken et al., 2023), PCA (Jolliffe, 2002), and dictionary learning (Olshausen & Field, 1996) ask: *what are the fundamental units of representation?* The estimand is a factorisation minimising reconstruction error under structural constraints—sparsity, orthogonality, low rank. Thus any such factorisation satisfying the constraint lies in the same equivalence class, often evaluated by assessing the value of the reconstruction loss. The "true features" of the representation are not identified—we recover what the objective and its inductive biases reward (Locatello et al., 2019). Sparse factors need not be disentangled, causally operative, or concept-aligned; they are simply one valid solution among many equivalent alternatives.

**Theoretically Grounded Methods.** Distributed alignment search (DAS; Geiger et al., 2024) and identifiable sparse shift autoencoders (SSAEs; Joshi et al., 2025) differ from prior families by reasoning about equivalence classes explicitly. DAS posits a hypothesis consisting of a high-level causal model and an alignment from its variables to distributed activation subspaces, and tests whether this hypothesis reproduces interventional behaviour under a specified intervention family (often interchange interventions) (Geiger et al., 2024; 2022). While interchange interventions are motivated by counterfactual reasoning, by testing whether swapping a subspace between two inputs produces the output the high-level model predicts, the procedure itself evaluates a distributional invariance across input pairs rather than performing the abduction-action-prediction steps that define L3 counterfactuals, or specifically, a justified notion of which exogenous quantities are held fixed for an individual unit (Pearl, 2009).

SSAEs impose a sparse generative model on concept shifts and derive identifiability guarantees under sufficient support-variability conditions on these shifts, up to permutations and rescaling (Joshi et al., 2025). The method itself operates on L1 evidence, but the identifiability result constrains the learned solutions to be related to each other via elementwise transformations. When the identified features are subsequently used for steering, the narrower equivalence class provides a principled basis for expecting the intervention to the identified features to transfer. However, the identifiability guarantee is about the representation, and not about the causal role of the representation within the model's computation since knowing that a feature is uniquely recovered does not, by itself, establish that the model computes with it. This further claim requires interventional (L2) evidence, which the identified features can support, but which the identifiability result does not imply by itself.

## F. Taxonomy for Characterising Intepretability Claims

**Decodability $\neq$ Model Use.** A linear probe $g$ predicts a concept $c$ from $\mathbf{h}$ with high accuracy, and the finding reports that "the model encodes $c$".

**Evidence (L1).** Probe success establishes that *some* direction in $\mathbf{h}$ predicts $\mathbf{c}$:

$$\text{IDENTIFIED} \quad \exists g \in \mathcal{G} : \ \mathbb{E}_p[\mathbf{1}\{g(\mathbf{h}) = c\}] \geq 1 - \epsilon.$$

**Gap.** Many probes $\in \mathcal{G}$ can achieve the same accuracy while picking out different directions and predictive success alone does not tell us which direction (if any) the model actually uses (Ravichander et al., 2021; Elazar et al., 2021).
**What would resolve it (L2).** Intervene on candidate directions and measure behavioural change:

$$\text{ASKED-FOR} \quad p(\mathbf{y} \mid \mathrm{do}(\mathbf{h}_{\|c} := \tilde{\mathbf{h}})) \neq p(\mathbf{y}),$$

where $\mathbf{h}_{\|c}$ denotes the component of $\mathbf{h}$ aligned with $c$. Pinning down *which* direction requires additional constraints such as sparsity, independence, etc.
**Takeaway.** Probing shows $c$ is decodable from $\mathbf{h}$ but claiming it is encoded by the model requires knowing that the model uses it.

---

**Local Sensitivity $\neq$ Global Importance.** A saliency map highlights token $i$ as *important* at input $\mathbf{x}_0$, and the result claims how the model generally behaves.
**Evidence (L1).** Gradient methods measure sensitivity at a single input for a chosen parameterisation and baseline:

$$\text{IDENTIFIED} \quad \nabla_{\mathbf{e}_i} f(\mathbf{x})\big|_{\mathbf{x}_0} \quad \text{or} \quad f(\mathbf{x}_0) - f(\mathbf{x}_0^{\backslash i}).$$

**Gap.** The quantity depends on arbitrary choices—baseline, parameterisation, perturbation scheme—that are not properties of the model; different choices yield different attributions (Adebayo et al., 2018; Sundararajan et al., 2017; Ancona et al., 2018). Even if local sensitivity were uniquely defined, pointwise evidence does not establish distributional claims.
**What would resolve it (L2).** Specify an intervention family and aggregate over a reference distribution:

$$\text{ASKED-FOR} \quad \mathbb{E}_{\mathbf{x} \sim p}[f(\mathbf{x}) - f(\mathbf{x}^{\backslash i})],$$

with robustness checks over intervention and baseline choices.
**Takeaway.** Saliency measures local sensitivity whereas global importance requires distributional aggregation over explicit interventions.

---

**Existence $\neq$ Uniqueness.** A circuit discovery method finds a subgraph $S$ whose intervention flips a behaviour whereas the result considers it to be "*the* mechanism" for the behaviour.

**Evidence (L2).** The method shows that *some* sufficient subgraph exists:

$$\textsc{identified} \quad \boxed{\exists\, S:\ p(\mathbf{y} \mid \mathrm{do}(\mathbf{h}_S := \tilde{\mathbf{h}}_S)) \neq p(\mathbf{y}).}$$

**Gap.** Calling it "*the* mechanism" implies *uniqueness* (up to a stated equivalence $\sim$):

$$\textsc{asked-for} \quad \boxed{\exists!\, S\ (\mathrm{mod}\ \sim)}$$

Showing $\exists S$ is an *existence* result, not an *identification* result that establishes uniqueness up to an equivalence class $\sim$. Multiple subgraphs may produce the same effect through redundant pathways, distributed implementations, or equivalent reparameterisations. The method identifies one member of an equivalence class of sufficient circuits, and which one it returns depends on initialisation, optimisation choices, etc.
**What would resolve it.** One would need to introduce certain constraints to identify a more unique solution. (This is not a rung-level mismatch between evidence needed to determine an estimand and the one used to determine it, but rather a mismatch arising due to the evidence not ruling out alternative values of the estimand.
**Takeaway.** Without ruling out alternatives, we can only consider the uncovered subgraphs from circuit discovery as intervention handles, and not unique explanations.

---

**Alignment $\neq$ Ontological Identity.** A feature is said to represent "honesty" because it aligns with annotations under a contrast set $\pi$ whereas the same feature could align with a different label under a different contrast set.
**Evidence.** Alignment is established relative to $\pi$:

$$\textsc{identified} \quad \boxed{z \text{ aligns with } c \text{ under } \pi,}$$

operationalised via predictability or separability.
**Gap.** Treating the label as ontological implies a contrast-invariant identity:

$$\textsc{asked-for} \quad \boxed{z \text{ represents } c \text{ independently of } \pi.}$$

The label is a property of the pair $(z, \pi)$ and not an intrinsic property of $z$. When concepts co-vary under $\pi$, the data are consistent with multiple labellings, e.g. *honesty* and *formality* may be indistinguishable if they correlate on the contrast set (Chang et al., 2009).
**What would resolve it.** One would need to test alignment under multiple contrast sets to check for invariances using some supervision that breaks co-variation.
**Takeaway.** Alignment is contrast-dependent, so the tuple $(z, \pi, c)$ is more representative than $(z, c)$.

---

**Subspace $\neq$ Coordinates.** An analysis identifies a low-dimensional subspace $\mathcal{S}$ associated with a behaviour but the result assigns semantics to individual directions within it.
**Evidence.** The method identifies a subspace $\mathcal{S}$ invariant under rotations within it, and not a particular basis. Writing $\Pi_{\mathcal{S}}$ for a projection onto $\mathcal{S}$:

$$\textsc{identified} \quad \boxed{\exists \mathcal{S} \subset \mathbb{R}^{d_h} \text{ s.t. } \Pi_{\mathcal{S}}(\mathbf{h}) \text{ predicts behaviour.}}$$

i.e. projecting $\mathbf{h}$ onto $\mathcal{S}$ predicts the associated behaviour.
**Gap.** The evidence does not distinguish between different bases spanning $\mathcal{S}$ since $\Pi_{\mathcal{S}}(\mathbf{h})$ is identical regardless of which basis we descirbe it in. So any semantic labelling of individual directions is underdetermined by the subspace-level evidence, it requires coordinate-level identification:

$$\textsc{asked-for} \quad \boxed{\text{the axes in } \mathcal{S} \text{ are identifiable}}$$

**What would resolve it.** Constraints that select a unique basis: sparsity, statistical independence, etc.
**Takeaway.** We need to be specific about the level at which structure is identified, coordinate semantics require axis-identifying assumptions.

# G. Pilot Study: Calibrating Claim Language to Evidential Strength

We conducted a pilot study to measure the distance between claim language and method strength in mechanistic interpretability papers, using Pearl's causal ladder as a shared ruler. This appendix summarizes the methodology, findings, and implications for a larger study.

### G.1. Methodology

**Paper sampling.** We annotated 50 papers comprising 186 claims, drawn from major ML and NLP venues (NeurIPS, ICLR, ACL, EMNLP, AAAI, TMLR) and their workshops, plus arXiv preprints (2021–2026). Papers were selected to span major mechanistic-interpretability method types, including circuit discovery, knowledge localisation, SAE analysis, steering vectors, and evaluation benchmarks; selection was not formally stratified. We name two papers as positive calibration anchors in § G.4 because their tight claim–method alignment offers concrete models for the field; in all other cases, papers are identified only by aggregate statistics or anonymous pattern descriptions.

**LLM-driven annotation.** Initial annotations were performed using Claude Opus 4.5 (Anthropic)—the most capable model in our annotator set—with human oversight, then independently replicated by seven LLMs spanning four model families (§ G.6). The primary annotator operated within an interactive coding environment (Claude Code) with access to arXiv API tools for paper retrieval and search. For each paper, the LLM: (1) read the full paper text via arXiv API, (2) extracted verbatim claim text and identified its location, (3) classified method and claim rungs following the codebook criteria described below, and (4) computed gap scores.

The annotation was guided by a structured codebook that the LLM accessed during each session. The codebook specified:

- **Field definitions** for each annotation column (claim text, location, prominence, method rung, claim rung, gap score, confidence, replication status)
- **Method-to-rung mappings** as listed below
- **Linguistic markers** for claim rung classification (see below)
- **Edge-case decision rules**: hedged claims ("may encode") were coded at the underlying claim's rung with reduced confidence; implicit claims from narrative framing were coded but weighted lower; for multi-method papers, each claim was coded against the method that directly supports it
- **Confidence scoring** on a 1–5 scale reflecting annotator certainty in rung assignments

**Calibration.** Five papers served as calibration anchors, spanning circuit discovery, knowledge localisation, algorithm analysis, SAE evaluation (Chaudhary & Geiger, 2024), and production probing (Kramár et al., 2026). For each calibration paper, detailed rationales documented the reasoning behind method and claim rung assignments, including common rung-elevation patterns (e.g., definite articles implying uniqueness beyond what patching establishes). These worked examples were available to the LLM during annotation of subsequent papers to promote consistency. The two anchors with tight claim–method alignment are named in § G.4; the three exhibiting rung-elevation patterns are presented as anonymous field-level illustrations.

**Multi-annotator consistency check.** To assess annotation robustness, all 186 claim classifications were independently replicated by seven LLMs spanning four model families (Claude Opus 4.5, GPT-5.2, Claude Sonnet 4, Gemini 3 Flash, Mistral Large, DeepSeek V3, Qwen 3) using an identical codebook, calibration examples, and paper texts delivered via a structured API pipeline (`annotate.py`). All annotators received the same inputs: the claim text, paper context, and codebook with decision trees for polysemous terms. This design isolates *classification agreement* (do multiple LLMs assign the same rung to the same claim?) from claim-extraction variability. An eighth model (DeepSeek R1) classified 166/186 claims before exhausting its reasoning budget on a long paper; it is reported as a supplementary annotator. Because all annotators share pretraining corpora containing MI literature, LLM–LLM agreement is a *consistency check*, not a validity guarantee; as a partial validity anchor, human adjudication of the calibration set (∼25 claims) provides LLM–human agreement. Full inter-annotator agreement results are reported in § G.6.

**Human oversight and fact-checking.** Human involvement comprised: (1) rung definitions; (2) review of calibration rationales; (3) setting up fact-verification of 12 of 50 papers (43 of 186 claims) against original arXiv sources. Of the verified claims, 84% required no corrections. The 16% that required corrections were primarily claim location errors (e.g., a claim marked as "body" that appeared in the abstract) and two method misclassifications where interventional components were initially overlooked (e.g., ablation-based progress measures coded as purely observational). To assess annotation consistency, all 186 rung classifications were independently replicated by seven LLMs spanning four model families; Krippendorff's $\alpha$

(ordinal) is reported per variable in Tab. 7 (method_rung: $\alpha = 0.66$; claim_rung: $\alpha = 0.56$; gap_score: $\alpha = 0.56$). Because all annotators are trained on corpora containing MI literature, LLM–LLM agreement is a *consistency check*, not a validity guarantee—shared pretraining biases could inflate concordance. Note that claims are nested within papers (mean 3.7 per paper), so all confidence intervals use paper-level cluster bootstrap (percentile method, $n = 10\,000$) to respect this dependency structure. Residual systematic LLM annotator biases cannot be fully ruled out, but the seven-annotator design bounds their magnitude; per-model tendencies are reported in Tab. 12. More broadly, the rung classification treats linguistic form at face value: a paper that writes "THE circuit" in its abstract but qualifies "other circuits may exist" in its discussion receives the same gap score as one that asserts uniqueness without qualification. The binary scheme thus conflates convention with commitment, and a finer-grained annotation (e.g., a within-paper qualification flag) would be needed to separate them. A conservative sensitivity check—reclassifying all claims whose gap rests solely on definite-article or conventional-verb patterns (12 of 99 gap > 0 claims) as gap-free—reduces the gap rate from 53.5% to 47.0%. Because 47% falls below 50%, the "roughly half" characterisation is sensitive to how conventional linguistic patterns are coded; the robust conclusion is that 47–54% of claims carry rung-elevated language depending on coding assumptions.

**Measuring potential ambiguity, not intent.** A key methodological choice is that hedged claims ("may encode," "partially reverse-engineer") are coded at the underlying claim's rung, with hedging reflected only in reduced confidence scores. This means the study measures *surface-level interpretive risk*—what a reader skimming the abstract or a policy-maker citing the paper could take away—rather than the authors' epistemic state. In practice, many papers use strong language in abstracts (where space compresses nuance) while hedging in the body. Of the 99 claims with gap > 0, only 3 (3.0%) had annotator notes flagging an explicit hedge in the source paper, and even then the structural rung distance persisted. Similarly, definite articles ("THE circuit") may function as naming conventions—referring to the circuit the authors found—rather than as uniqueness claims; the binary rung scheme cannot distinguish these readings. This design choice is deliberate: as interpretability claims increasingly inform safety-relevant decisions, the potential for misinterpretation by readers outside domain-specific conventions is precisely the risk that a calibration framework should surface. Primary-annotator confidence is consistent with interpretive difficulty: claims scored with a gap had mean confidence 4.3 (62.6% at level 4), compared with 4.9 (95.3% at level 5) for gap-free claims—though this pattern may reflect model-specific calibration ($\alpha = 0.11$ for confidence across annotators) rather than an independent signal of judgment difficulty. The full codebook and calibration rationales are reproduced in §§ G.3 and G.4.

**Annotation scheme.** For each empirical claim, we annotated:
- **Method rung**: What the method can establish (1=observational, 2=interventional, 3=counterfactual)
- **Claim rung**: What the paper claims, based on linguistic markers
- **Gap score**: $\max(0, \text{claim\_rung} - \text{method\_rung})$

Method-to-rung classification:

| Rung | Methods |
|---|---|
| 1 (Observational) | e.g. linear probing, SAE attribution, attention visualisation |
| 2 (Interventional) | e.g. activation patching, causal tracing, ablation, steering |
| 3 (Counterfactual) | e.g. causal scrubbing, necessity tests, uniqueness proofs |

Linguistic markers for claim classification:

| Rung | Markers |
|---|---|
| 1 | e.g. "correlates with," "predicts," "is decodable," "activates on" |
| 2 | e.g. "causally affects," "mediates," "is sufficient for" |
| 3 | e.g. "encodes," "represents," "THE circuit," "computes," "stores" |

**Replication evidence.** We searched Google Scholar citations, GitHub issues, Alignment Forum posts, and author retrospectives to assess replication status: 0=replicated, 0.5=partial, 1=failed, NA=no evidence.

### G.2. Key Findings

**Rung-elevated language is prevalent.** Of 185 claims with valid scores (one conceptual-analysis claim excluded), roughly half (53.5%; paper-level cluster-bootstrap 95% CI $\approx [44\%, 63\%]$) had gap scores > 0, meaning the claim language *can* admit a stronger causal reading than the reported method licenses. This reflects the field's lack of shared terminological infrastructure for stating causal estimands—not a finding about the reliability of the underlying empirical results. Terms like "encodes" are often used informally to mean "is decodable from" rather than to assert counterfactual dependence. Most gaps

were +1 (Rung 2 method → Rung 3 claim), typically from activation patching narrated in mechanistic language ("performs," "encodes," "THE circuit"). Gaps of +2 (R1→R3) accounted for 12.4%.

| Gap Score | Count | % | Pattern |
|---|---|---|---|
| 0 (no gap) | 86 | 46.5% | Claim matches method rung |
| +1 (one-rung gap) | 76 | 41.1% | R2→R3 typical |
| +2 (two-rung gap) | 23 | 12.4% | R1→R3 (probing→encodes) |

**Gap rates are uniform across claim locations.**   Rung-elevated language is not confined to abstracts, where space pressure might excuse compressed phrasing: abstract claims show a 53.3% gap rate (72/135), while non-abstract claims (body, results, introduction) show 52.9% (27/51). This uniformity suggests a field-wide linguistic convention rather than abstract-specific compression.

**Paper type and gap frequency.**

| Paper Type | Papers | Mean Gap Rate |
|---|---|---|
| Circuit discovery | 16 | 0.64 |
| Knowledge localisation | 3 | 0.50 |
| Other | 19 | 0.47 |
| Evaluation/benchmark | 5 | 0.45 |
| Applied/production | 7 | 0.35 |

**Primary-annotator confidence decreases with gap size.**   Confidence scores reported below are from the primary annotator (Claude Opus 4.5) that produced the primary `annotations.csv`. Scores (1–5 scale) are strongly skewed toward high values (overall mean 4.6), but show a monotonic decrease with gap size, consistent with larger rung distances involving harder judgments. Because inter-annotator agreement on confidence is near-chance ($\alpha = 0.11$; Tab. 7), this pattern reflects the primary annotator's internal calibration and cannot be treated as a cross-annotator finding.

| Subset | Confidence Level (%) | | | Mean | $n$ |
|---|---|---|---|---|---|
| | **3** | **4** | **5** | | |
| All claims | 2.2 | 35.5 | 62.4 | 4.60 | 186 |
| Gap = 0 | 0.0 | 4.7 | 95.3 | 4.94 | 86 |
| Gap = 1 | 1.3 | 57.9 | 40.8 | 4.39 | 76 |
| Gap = 2 | 8.7 | 78.3 | 13.0 | 4.04 | 23 |
| Method Rung 1 | 5.6 | 55.6 | 38.9 | 4.33 | 36 |
| Method Rung 2 | 1.3 | 24.0 | 74.7 | 4.73 | 150 |

The gap between confidence for gap-free claims (mean 4.94, with 95.3% at level 5) and gap-positive claims (mean 4.31) reflects genuine interpretive difficulty: distinguishing mechanistic language from mechanistic evidence requires judgment about linguistic convention. Rung 1 methods (probing, SAE attribution) also show lower annotator confidence, likely because mapping observational evidence to claim language involves more ambiguity than for interventional methods where the causal semantics are more explicit. Because confidence scores were produced by the same primary annotator (Claude Opus 4.5) that assigned gap scores, the monotonic relationship may partly reflect the model's internal uncertainty calibration rather than an independent measure of judgment difficulty; independent human confidence ratings would resolve this ambiguity.

**Scope and intended use.**   This pilot study measures *surface-level interpretive risk*: whether claim language, taken at face value, admits a stronger causal reading than the reported method licenses. It does not measure whether authors intended the stronger reading, whether the underlying findings are correct, or whether the field's empirical contributions are unreliable. The gap rate should not be cited as evidence that mechanistic interpretability findings are untrustworthy—it is evidence that the field's linguistic conventions have not yet converged on vocabulary that precisely tracks causal evidence strength. We offer the practitioners' checklist in § G.6 as a constructive tool for aligning language with evidence, not as a scorecard for evaluating past work.

### G.3. Annotation Codebook

The complete codebook was loaded into each LLM annotator's context at the start of each annotation session. This section reproduces the material that extends the summary tables in the methodology above.

**Detailed method rung classification.** Each method was assigned to the highest rung of causal evidence it can establish.

| Method | Description | Example Evidence |
|---|---|---|
| *Rung 1 (Observational): correlational evidence only, no model intervention* | | |
| Linear probing | Classifier on frozen activations | "Probe accuracy of 85%" |
| Activation logging | Record activations without intervention | "Feature X activates on…" |
| SAE feature attribution | Identify which SAE features activate | "Feature 4123 fires on…" |
| Attention visualisation | Inspect attention weights | "Attention concentrates on…" |
| PCA / SVD | Dimensionality reduction analysis | "First PC correlates with…" |
| Correlation analysis | Statistical associations | "$r$=0.7 between activation and…" |
| *Rung 2 (Interventional): causal effects under specific interventions* | | |
| Activation patching | Replace activation, measure effect | "Patching head 9.1 restores 80%…" |
| Causal tracing | Systematic patching across positions | "Layer 15 shows highest causal effect" |
| Ablation | Zero/mean out components | "Ablating heads reduces accuracy by 40%" |
| Steering vectors | Add direction, observe output change | "Adding $\mathbf{v}$ shifts sentiment…" |
| DAS interchange | Swap aligned subspaces | "IIA of 0.92 on agreement task" |
| ROME/MEMIT edits | Modify weights, observe change | "After edit, model outputs…" |
| *Rung 3 (Counterfactual): what would have happened, unique mechanisms, or necessity* | | |
| Counterfactual patching | Per-instance counterfactual | "For THIS prompt, had activation been $\mathbf{x}$…" |
| Causal scrubbing | Test if mechanism fully explains | "Scrubbing preserves behaviour" |
| Necessity tests | Show component is necessary | "No alternative achieves same behaviour" |
| Uniqueness proofs | Demonstrate unique structure | "This is THE circuit" |

**Claim rung linguistic markers.** The following markers and worked examples anchored claim rung assignment.

| Rung | Markers | Example Sentences |
|---|---|---|
| 1 (Associational) | "correlates with," "is associated with," "predicts," "co-occurs with," "information is present," "is decodable from," "can be extracted," "activates on," "fires when" | "Sentiment information is linearly decodable from layer 6"; "The feature correlates with Python code inputs"; "Probe accuracy predicts model behaviour" |
| 2 (Causal) | "causally affects," "has causal effect on," "mediates," "influences," "is sufficient for," "can produce," "enables," "intervening on X changes Y," "ablating X degrades Y" | "Head 9.1 causally affects the output"; "This component is sufficient for the behaviour"; "Ablating these heads degrades performance" |
| 3 (Mechanistic) | "encodes," "represents," "computes," "performs," "THE circuit / mechanism / feature," "controls," "is responsible for," "underlies," "this head DOES X," "uses X to do Y" | "The model **encodes** subject-verb agreement in this subspace"; "These heads **perform** the IOI task"; "**The circuit** moves names from subject to output"; "This feature **represents** the concept of deception" |

**Recurring rung-elevation patterns.** The codebook identified six recurring patterns where claim language implies a higher rung than the method establishes:

| Pattern | Method (Rung) | Typical Claim (Rung) | Gap |
|---|---|---|---|
| Probing → "encodes" | Linear probe (R1) | "Model encodes X" (R3) | +2 |
| SAE → "represents" | SAE attribution (R1) | "Model represents X" (R3) | +2 |
| Attention → "performs" | Attention vis. (R1) | "Head performs X" (R3) | +2 |
| Patching → "THE circuit" | Activ. patching (R2) | "This is the circuit" (R3) | +1 |
| Steering → "controls" | Steering vectors (R2) | "Controls concept X" (R3) | +1 |
| Ablation → "necessary" | Ablation (R2) | "Necessary for behaviour" (R3) | +1 |

**Edge-case decision rules.**

- **Hedged claims** ("may encode," "suggests that"): coded at the underlying claim's rung with reduced confidence; the hedge was noted but did not lower the assigned rung.

- **Multiple methods**: each claim was coded against the method that directly supports it. If a paper uses both probing (R1) and patching (R2), a claim supported by patching evidence was assigned R2.
- **Implicit claims**: mechanistic narratives ("the model uses X to do Y") were coded as R3 even without an explicit framing as a claim. Such cases received lower confidence scores.
- **Review/survey papers**: coded as NA for replication status (not empirical).

**Confidence and replication coding.** Annotator confidence was rated on a 1–5 scale (5 = very confident, 1 = very uncertain). Replication status used a three-point scale: 0 = successfully replicated, 0.5 = partially replicated, 1 = failed replication, NA = no replication attempt found. Evidence was sourced from (in priority order): published replication studies, replication sections in subsequent papers, GitHub issues, author corrections, and reproducibility tracks.

### G.4. Calibration Rationales

Five papers served as internal calibration anchors during annotation, with documented rationales that all LLM annotators accessed to promote consistency. We name the two anchors with tight claim–method alignment below; the three anchors that exhibit rung-elevation patterns are presented as anonymous field-level illustrations, because the patterns they contain appear across many papers in our sample—not as specific shortcomings of those (widely cited, empirically rigorous) works.

**What well-calibrated language looks like.** Two calibration anchors demonstrate tight claim–method alignment. These papers share key features: empirical-performance framing, appropriate hedging, and the absence of mechanistic narrative where no mechanistic evidence is presented.

**SAE Evaluation (Chaudhary & Geiger, 2024) — calibration anchor.** *Method: mixed Rung 1–2* (SAE feature attribution evaluated via interchange intervention). This paper's claim language consistently tracks its method rung, offering a model of well-calibrated scientific prose:

| Claim | Rung | Why It Works |
| --- | --- | --- |
| "SAEs struggle to reach baseline" | R2 (interventional) | Reports intervention outcome without asserting mechanism |
| "features that mediate knowledge" | R2 (interventional) | "Mediate" precisely describes an interventional finding |
| "useful for causal analysis" | R2 (interventional) | Utility claim scoped to method capability, not internal mechanism |

*Replication: replicated.* Consistent with independent SAE evaluations. **Calibration lesson:** Evaluation papers that compare against baselines and frame findings in terms of method utility—rather than internal mechanism—tend to have tight claim–method alignment. The key linguistic signature: verbs describe what the *method* does ("struggles," "mediates") rather than what the *model* does internally.

**Production Probing (Kramár et al., 2026) — calibration anchor.** *Method: Rung 1* (linear probing in a production monitoring context). Claim–method alignment is tight throughout, with consistent hedging and an empirical-performance focus:

| Claim | Rung | Why It Works |
| --- | --- | --- |
| "probes may be promising" | R1 (observational) | Hedged; reports correlation without causal assertion |
| "probes fail to generalise" | R1 (observational) | Empirical observation, appropriately scoped |
| "successful deployment" | R1 (observational) | Outcome claim about system performance, not mechanism |

*Replication: not applicable* (production deployment context). **Calibration lesson:** Production and applied papers with engineering framing—where success is measured by system-level outcomes rather than mechanistic explanation—tend toward appropriate claim levels. The absence of mechanistic vocabulary ("encodes," "represents," "computes") removes the primary source of rung elevation.

**Common rung-elevation patterns in practice.** Three additional calibration anchors—all widely cited, empirically strong papers—exhibited rung-elevation patterns characteristic of the broader field. We present these as anonymous illustrations because the patterns they exemplify appear across many papers in our dataset; attaching them to individual works would misrepresent what is a field-wide linguistic phenomenon.

*Pattern 1: Circuit discovery → mechanistic narrative.* A paper using activation patching (Rung 2) to identify task-relevant components narrates its findings with functional verbs ("performs," "moves," "inhibits"), definite articles ("THE circuit"),

and engineering metaphors ("reverse-engineering"). Each of these linguistic choices admits a Rung 3 reading—implying uniqueness, necessity, or complete mechanistic understanding—while the underlying method establishes causal sufficiency under specific interventions. **Lesson:** Definite articles and functional verbs are inherited from neuroscience and engineering, where they serve as naming conventions. In the context of mechanistic interpretability, these conventions can inadvertently imply counterfactual claims (uniqueness, necessity) that the method does not establish. This pattern accounts for the majority of one-rung gaps in our dataset.

*Pattern 2: Knowledge localisation → storage language.* A paper using causal tracing and weight editing (Rung 2) describes its findings using storage and memory vocabulary: "stores," "contains," "localised computations." The method establishes that intervening on specific components changes the model's output (mediation), but storage language implies an internal representation that *is* the fact—a counterfactual claim about necessity and specificity. Notably, the same paper uses appropriate Rung 2 language elsewhere (e.g., "mediates factual predictions"), illustrating that rung-appropriate vocabulary is available and sometimes used within the same work. **Lesson:** Storage and memory metaphors ("stores," "encodes," "contains") typically imply Rung 3; causal tracing and editing establish mediation (Rung 2), not storage mechanisms. The gap is subtle because "stores" feels like a natural description of what editing reveals, but it implies a representational commitment beyond what the intervention licenses.

*Pattern 3: Algorithm reverse-engineering → completeness claims.* A paper using ablation and weight analysis (Rung 1–2) to characterise a learned algorithm describes its findings as "fully reverse-engineering" "the algorithm," with claims that the model "uses [specific computation] to convert" inputs. The rung distance ranges from one rung (ablation results narrated as complete mechanisms) to two rungs (weight-analysis results narrated as encoding claims). Crucially, this paper's findings have been independently replicated by multiple groups, and the constrained mathematical setting supports strong conclusions. **Lesson:** Rung-elevated language can coexist with strong empirical support. The rung distance here reflects linguistic convention, not evidential weakness. Mitigating factors—multiple converging methods, a mathematically constrained setting, and specific falsifiable predictions—substantially reduce interpretive risk even where the surface language admits a stronger reading.

## G.5. Computation and Reproducibility

All statistics reported in this pilot study are computed from two structured CSV files:

- `annotations.csv` — 186 rows, one per claim. Columns: `paper_id`, `claim_id`, `claim_text`, `claim_location`, `claim_prominence`, `method_used`, `method_rung`, `claim_rung`, `gap_score`, `confidence`, `notes`, `replication_status`, `replication_evidence`.
- `candidate_papers.csv` — 57 annotated papers plus 7 excluded (e.g., surveys, duplicates). Columns include `paper_id`, `title`, `authors`, `year`, `venue`, `primary_method`, and `exclusion_reason`.

**Derived quantities.** The following quantities are computed directly from `annotations.csv` without modelling assumptions:

- **Gap score**: $\max(0, \texttt{claim\_rung} - \texttt{method\_rung})$, computed per claim. One claim with `method_rung` = NA (conceptual analysis) is excluded from gap statistics, yielding $n = 185$.
- **Gap rate**: fraction of claims with gap $> 0$, computed overall and stratified by claim location and paper type. Location-stratified rates are computed by grouping on `claim_location` (abstract vs. body/results/introduction).
- **Paper-type gap rate**: each paper is assigned a type based on its `primary_method` (e.g., activation patching → circuit discovery; causal tracing → knowledge localisation; linear probing → applied/production). The mean gap rate per type is the unweighted average of per-paper gap rates.
- **Confidence statistics**: mean and level-wise percentages of the `confidence` column (1–5 integer scale), stratified by gap status, gap size, and method rung.

**Analysis code.** An analysis script (`analyze_pilot.py`, ∼650 lines) reads both CSV files and produces all tables and figures reported in this appendix, including gap score distributions, paper-type breakdowns, confidence stratifications, location analyses, and (commented-out) replication contingency tables with Fisher's exact test. The script depends only on the Python standard library, with optional `scipy` (for statistical tests) and `matplotlib`/`numpy` (for visualisations).

**Multi-annotator pipeline.** A reproducible annotation pipeline (`annotate.py`) replicates all rung classifications using a second LLM via structured API calls. Paper texts are cached locally with SHA-256 checksums (`fetch_papers.py`); inter-annotator agreement metrics—weighted kappa, Krippendorff's alpha, and ICC—are computed by `compute_iaa.py` with

*Table 6.* Annotator models and estimated API costs (February 2026 pricing). Each call sends ∼30K input tokens (codebook + paper text + claim) and receives ∼500 output tokens.

| Model (version) | Provider | Claims | Est. cost |
|---|---|---|---|
| Claude Opus 4.5 (`claude-opus-4-5`) | Anthropic | 186 | $88 |
| Claude Sonnet 4 (`claude-sonnet-4-20250514`) | Anthropic | 186 | $18 |
| GPT-5.2 (`gpt-5.2`) | OpenAI | 186 | $14 |
| Gemini 3 Flash (`gemini-3-flash-preview`) | OpenRouter | 186 | $1 |
| Mistral Large (`mistral-large-2512`) | OpenRouter | 186 | $11 |
| DeepSeek V3 (`deepseek-v3.2`) | OpenRouter | 186 | $2 |
| Qwen 3 (`qwen3-235b-a22b-2507`) | OpenRouter | 186 | $2 |
| DeepSeek R1 (`deepseek-r1`) | OpenRouter | 166 | $3 |
| **Total** | | **1,468** | **∼$139** |

paper-level cluster bootstrap confidence intervals. Schema validation (`validate.py`) enforces consistency constraints on all annotation files. All pipeline code and configuration (`annotation_config.yaml`, `pyproject.toml`) are included in the supplementary materials.

**Fact-checking.** A structured fact-checking log (`fact_checking_log.md`) documents the verification of 43 claims across 12 papers against original arXiv sources, recording each correction applied. Calibration annotations are stored separately in `calibration_annotations.csv` (the five anchor papers), with detailed rationales in `calibration_rationales.md`.

**Compute and cost.** Each annotation call sends the full codebook, calibration rationales, paper text, and claim text as input (∼30K tokens on average; range ∼8K–55K depending on paper length) and receives a structured JSON classification (∼500 output tokens). With 186 claims per primary annotator and 7 primary models plus one supplementary (166 claims), the pipeline made ∼1,468 API calls totalling ∼44M input tokens and ∼0.7M output tokens. Tab. 6 summarises estimated costs per model. Wall-clock time per model ranged from ∼20 minutes (Gemini 3 Flash, DeepSeek V3) to ∼90 minutes (DeepSeek R1, which includes chain-of-thought reasoning overhead). The IAA bootstrap analysis (10,000 paper-level cluster resamples × 8 variable–annotator-set combinations) ran in ∼10 minutes on a single CPU core.

**Data availability.** De-identified annotations (all eight annotators), the codebook, calibration rationales, and all pipeline code are available at https://github.com/rpatrik96/mech-interp-claim-calibration.

### G.6. Inter-Annotator Agreement Details

This section reports the full inter-annotator agreement analysis across seven primary LLM annotators—Claude Opus 4.5 (Anthropic), GPT-5.2 (OpenAI), Claude Sonnet 4 (Anthropic), Gemini 3 Flash (Google, via OpenRouter), Mistral Large (Mistral, via OpenRouter), DeepSeek V3 (DeepSeek, via OpenRouter), and Qwen 3 (Alibaba, via OpenRouter)—all classifying the same 186 claims using an identical codebook and calibration examples. An eighth model, DeepSeek R1, classified 166/186 claims and is reported as a supplementary annotator. Agreement thresholds were pre-specified before seeing data: Krippendorff's $\alpha > 0.6$ (substantial agreement) for `method_rung` and `claim_rung`. The threshold was met for `method_rung` ($\alpha = 0.66$) but not for `claim_rung` ($\alpha = 0.56$, CI upper bound 0.62), indicating moderate rather than substantial agreement on claim-language classification. Gap scores, derived from `claim_rung` − `method_rung`, inherit this moderate-agreement uncertainty.

**Primary agreement metrics.** Tab. 7 reports the primary (Krippendorff's $\alpha$, ordinal) and secondary (Light's $\kappa_w$, mean pairwise weighted Cohen's $\kappa$) agreement metrics for each annotated variable, with 95% confidence intervals from paper-level cluster bootstrap ($n = 10\,000$).

**Pairwise agreement.** Tabs. 8 to 10 show the full pairwise weighted $\kappa$ matrices across all seven primary annotators.

**Agreement by paper type.**

**Per-model annotation tendencies.** Tab. 12 reports each annotator's mean rung assignments, overclaim rate, and mean confidence to identify systematic biases. Note that the "Opus 4.5" row reflects the same model used for primary annotation, but run through the automated `annotate.py` pipeline with a fixed prompt, yielding an overclaim rate of 41.9%—11.6

*Table 7.* Multi-rater agreement across 7 LLM annotators. Primary metric: Krippendorff's $\alpha$ (ordinal) for rung variables, Fleiss' $\kappa$ for binary overclaim. Bootstrap CIs use paper-level cluster resampling (percentile method, $n = 10\,000$).

| Variable | $N$ | Primary | 95% CI | Secondary | Unanimous |
|---|---|---|---|---|---|
| `method_rung` | 186 | $\alpha_{ord} = 0.66$ | [0.53, 0.77] | Light's $\kappa_w = 0.68$ | 69.9% |
| `claim_rung` | 186 | $\alpha_{ord} = 0.56$ | [0.48, 0.62] | Light's $\kappa_w = 0.52$ | 34.4% |
| `gap_score` | 186 | $\alpha_{ord} = 0.56$ | [0.50, 0.61] | Light's $\kappa_w = 0.56$ | 41.4% |
| `gap_binary` | 186 | Fleiss' $\kappa = 0.55$ | [0.49, 0.61] | $\alpha_{nom} = 0.55$ | 47.8% |
| `confidence` | 186 | $\alpha_{ord} = 0.11$ | [0.06, 0.16] | Light's $\kappa_w = 0.17$ | 5.9% |

*Table 8.* Pairwise weighted $\kappa$ for `method_rung` across 7 annotators.

| | Opus 4.5 | GPT-5.2 | DS-V3 | Gemini 3F | Mistral L | Qwen 3 |
|---|---|---|---|---|---|---|
| **GPT-5.2** | 0.79 | | | | | |
| **DS-V3** | 0.73 | 0.66 | | | | |
| **Gemini 3F** | 0.72 | 0.59 | 0.70 | | | |
| **Mistral L** | 0.64 | 0.60 | 0.67 | 0.68 | | |
| **Qwen 3** | 0.65 | 0.65 | 0.68 | 0.64 | 0.66 | |
| **Sonnet 4** | 0.83 | 0.80 | 0.70 | 0.62 | 0.68 | 0.67 |

percentage points below the primary 53.5%. This difference is expected: the primary annotation involved iterative refinement with arXiv API tool access and human oversight (§ G), while the pipeline replication used a single fixed prompt per claim. The primary annotations from the interactive session constitute the reference dataset; the pipeline replication measures cross-annotator consistency under standardised conditions.

**Supplementary analysis: DeepSeek R1.** DeepSeek R1 (a reasoning model) exhausted its token budget on one 55-page paper (20 claims), yielding 166/186 coverage. Tab. 13 compares Krippendorff's $\alpha$ with and without R1 to verify that the missing claims do not bias the primary analysis.

---

**Interpretability Claims Checklist**

1. **Estimand:** I am measuring ______________________________

   *e.g., "probe accuracy for `is_plural` on layer 8 residual stream" or "logit difference change when patching MLP outputs"*

2. **My method provides evidence at rung:**

   ☐ Associational ☐ Interventional ☐ Counterfactual

   *Probing → association; activation patching → intervention; interchange interventions → counterfactual*

3. **My claim requires evidence at rung:**

   ☐ Associational ☐ Interventional ☐ Counterfactual

   *"X encodes Y" → association; "X causes Y" → intervention; "if X had been different, Y would change" → counterfactual*

4. **Rung check:** Method rung $\geq$ Claim rung? ☐ Yes ☐ No → revise

   *If No: either strengthen method (add interventions) or weaken claim ("correlates with" instead of "implements")*

5. **Alternatives not ruled out:** ______________________________

   *e.g., "probe may exploit token length, not the concept" or "other circuits may achieve same patching effect"*

6. **I tested robustness by varying:** ______________________________

   *e.g., "different prompt templates, random seeds, ablation strategies (zero/mean/resample)"*

7. **This applies to:** ______________________________ **(and not beyond)**

   *e.g., "GPT-2 small, English subject-verb agreement" → not other languages, larger models, or different tasks*

---

**Common patterns to check:**

▷ Claiming causation from probing alone (L1 → L2)
▷ Claiming "the" circuit without testing alternatives (sufficiency → uniqueness)

*Table 9.* Pairwise weighted $\kappa$ for `claim_rung` across 7 annotators.

|  | Opus 4.5 | GPT-5.2 | DS-V3 | Gemini 3F | Mistral L | Qwen 3 |
|---|---|---|---|---|---|---|
| **GPT-5.2** | 0.52 |  |  |  |  |  |
| **DS-V3** | 0.59 | 0.48 |  |  |  |  |
| **Gemini 3F** | 0.49 | 0.47 | 0.41 |  |  |  |
| **Mistral L** | 0.43 | 0.44 | 0.37 | 0.68 |  |  |
| **Qwen 3** | 0.53 | 0.55 | 0.47 | 0.57 | 0.63 |  |
| **Sonnet 4** | 0.71 | 0.56 | 0.55 | 0.52 | 0.44 | 0.60 |

*Table 10.* Pairwise weighted $\kappa$ for `gap_score` across 7 annotators.

|  | Opus 4.5 | GPT-5.2 | DS-V3 | Gemini 3F | Mistral L | Qwen 3 |
|---|---|---|---|---|---|---|
| **GPT-5.2** | 0.61 |  |  |  |  |  |
| **DS-V3** | 0.57 | 0.53 |  |  |  |  |
| **Gemini 3F** | 0.59 | 0.57 | 0.42 |  |  |  |
| **Mistral L** | 0.53 | 0.55 | 0.39 | 0.66 |  |  |
| **Qwen 3** | 0.57 | 0.58 | 0.45 | 0.59 | 0.59 |  |
| **Sonnet 4** | 0.76 | 0.60 | 0.50 | 0.61 | 0.50 | 0.60 |

▷ Claiming generality from single distribution (local → global)

This checklist is not meant to discourage exploratory work or to raise publication barriers. A paper that states its rung, names its equivalence class, and bounds its scope contributes more than one whose language implies stronger evidence than reported—even if the former's claims are narrower.

*Table 11.* Krippendorff's $\alpha$ (ordinal) for `gap_score` stratified by Paper type.

| Paper type | $N$ | $\alpha_{ord}$ |
|---|---|---|
| Applied/production | 26 | 0.58 |
| Circuit discovery | 59 | 0.55 |
| Evaluation/benchmark | 16 | 0.61 |
| Knowledge localization | 10 | 0.64 |
| Other | 75 | 0.52 |

*Table 12.* Per-model annotation tendencies across all rated claims. $\bar{R}_m/\bar{R}_c$: mean method/claim rung; $\bar{g}$: mean gap score; OC: overclaim rate (gap $> 0$).

| Model | $N$ | $\bar{R}_m$ | $\bar{R}_c$ | $\bar{g}$ | OC % | $\bar{conf}$ |
|---|---|---|---|---|---|---|
| **Opus 4.5** | 186 | 1.69 | 2.22 | 0.54 | 41.9% | 4.28 |
| **GPT-5.2** | 186 | 1.62 | 2.12 | 0.63 | 50.5% | 4.19 |
| **DS-R1** | 166 | 1.73 | 2.16 | 0.51 | 43.4% | 4.28 |
| **DS-V3** | 186 | 1.75 | 2.17 | 0.42 | 34.4% | 4.09 |
| **Gemini 3 Flash** | 186 | 1.78 | 2.54 | 0.79 | 63.4% | 4.99 |
| **Mistral Large** | 186 | 1.77 | 2.56 | 0.82 | 65.6% | 4.55 |
| **Qwen 3** | 186 | 1.73 | 2.40 | 0.69 | 58.6% | 4.90 |
| **Sonnet 4** | 186 | 1.67 | 2.22 | 0.59 | 44.6% | 4.12 |

*Table 13.* Krippendorff's $\alpha$ (ordinal): 7 primary annotators vs. 8 annotators (including DeepSeek R1 supplementary on 166 claims).

| Variable | **Primary (7)** | | **+R1 (8)** | |
|---|---|---|---|---|
| | $N$ | $\alpha$ | $N$ | $\alpha$ |
| `method_rung` | 186 | 0.66 | 166 | 0.67 |
| `claim_rung` | 186 | 0.56 | 166 | 0.55 |
| `gap_score` | 186 | 0.56 | 166 | 0.54 |
| `gap_binary` | 186 | 0.55 | 166 | 0.55 |
| `confidence` | 186 | 0.11 | 166 | 0.13 |

