# OpenReview forum: "Position: Causality Is Key for Interpretability Claims to Generalise"
_ICML.cc/2026/Position_Paper_Track — ICML 2026 Position Paper Track regular_

### Official Review · Reviewer_sPgE · 2026-03-05

**Significance:** 4
**Argument Clarity:** 2
**Rating:** 3
**Confidence:** 5

**Questions:**

Please see my comments in the weakness section.

**Alternative Views Section:**

Yes

**Compliance With Llm Reviewing Policy A Conservative:**

Affirmed.

**Discussion Potential:**

3

**Final Justification:**

my main concerns remain only partially resolved. In particular, I still find the causal assumptions and abstraction mapping insufficiently clear in the main presentation, and I remain unconvinced that the paper yet provides enough practical guidance or empirical support for how this framework would improve interpretability work in practice. For this reason, my overall assessment remains unchanged.

**Paper Summary:**

The paper argues that many interpretability claims about LLMs are stronger than the evidence supporting them. It proposes using causal inference (Pearl’s causal hierarchy, and the field of identification) to formalize what interpretability experiments actually prove. The authors show that many studies provide only associational or sometimes interventional evidence, yet make stronger causal claims such as identifying mechanisms. They introduce a framework to diagnose this mismatch and argue that interpretability research should define causal estimands and identification assumptions so that claims match the evidence and generalize across settings.

**Position:**

Yes

**Position In Title:**

Yes

**Related Work:**

3

**Strengths And Weaknesses:**

**Strengths**

* Clear conceptual framing: The paper provides a useful perspective by framing interpretability claims through causal inference and identifiability.

* Important problem: It highlights a real issue in the literature where correlational or interventional evidence is sometimes interpreted as demonstrating mechanisms.

* Useful vocabulary: Concepts such as rung mismatch and identification gap could help make interpretability claims more precise.

**Weaknesses**

* Unclear causal assumptions: The paper does not specify what causal variables or structural assumptions should be used when modeling neural networks or LLMs, which makes the framework difficult to operationalize. (also my understanding is that causal assumptions are the key to reach identification, which is often build on domain knowledge (or from outside sources), how and where this information can be framed and extracted in LLM to me is the real challenge)

* Limited empirical support: The work is largely conceptual and does not demonstrate how adopting the proposed framework would improve interpretability methods in practice.

* Ambiguity in causal abstraction: Interpretability methods operate at different levels (neurons, circuits, features), but the paper does not address how these map to causal variables.

**Support:**

2

---

> ### Author Rebuttal · Authors · 2026-03-31
>
> We thank you for your careful review. Your confirmation that the paper highlights *a real issue where correlational or interventional evidence is sometimes interpreted as demonstrating mechanism* validates our core motivation. We are encouraged that you find rung mismatch and identification gap *useful vocabulary* that *could help make interpretability claims more precise, which is exactly a contribution we hope to make, shared by both reviewers J2gy and FGv2. We address your concerns below.
>
> > Unclear causal assumptions:
>
> The three different case studies in the main text provide specific worked out analyses involving different causal variables, ranging from neurons to input prompts, model weights, and learned concepts. We explicitly discuss these in Appendix A.1, which maps intervenable objects to different methods (Table 3). Appendix A.2 lists four feature operationalisations (neurons, directions, dictionary elements, circuits) and discusses their implications. Table 4 provides various examples of all three levels of causal queries framed through different variables in an LLM. Appendix F provides further case studies that demonstrate the framework at each level. Thank you for pointing out that our formulation was unclear. We will make these parts more prominent.  We welcome any additional pointers on how we can make it clearer “how and where this information can be framed and extracted in LLM”.
>
> > Limited empirical support
>
> We agree the paper has limited empirical evidence. We provide a pilot study (Sec 2.3) across 50 papers and show 53% _potential_ overclaiming to directly validate any implications of the claim mismatch we present in the paper. We also cite several examples of papers that retroactively predict known failures through the case studies. Lastly, we mention DAS and SSAEs as examples of causally grounded methods (Sec 2.2) for LLMs. A full prospective study is beyond a position paper's scope but is outlined in the Call to Action (Sec 4).
>
> > Ambiguity in causal abstraction
>
> Thanks for pinpointing this ambiguity, we will clarify our current discussion, which is as follows. The paper's structure traces the mapping explicitly: activations $\mathbf{a}^{(l)}$ are the fundamental unit but are not aligned to interpretable axes (Sec 2, Notation). Features are recovered by learning a map $\phi$ via regularised methods. Circuits are causal models over neurons or over features (Case Study I). The three case studies demonstrate the framework at the direction level (steering), dictionary level (SAEs), and circuit level (patching). Appendices A.1--A.2, F address this question explicitly to model causal variables from different levels and components of an LLM. We will make this clearer in the main text, thank you for the recommendation.

---

> > ### Author Rebuttal · Reviewer_sPgE · 2026-03-31
> >
> > Thank you for the rebuttal. The response clarified several aspects of the paper, especially the intended role of causal assumptions, the abstraction levels involved, and where some of the operational details are currently located. I agree the paper raises an important issue and provides a potentially useful vocabulary for making interpretability claims more precise.
> >
> > That said, my main concerns remain only partially resolved. In particular, I still find the causal assumptions and abstraction mapping insufficiently clear in the main presentation, and I remain unconvinced that the paper yet provides enough practical guidance or empirical support for how this framework would improve interpretability work in practice. For this reason, my overall assessment remains unchanged.

---

### Official Review · Reviewer_FGv2 · 2026-03-05

**Significance:** 3
**Argument Clarity:** 4
**Rating:** 5
**Confidence:** 2

**Questions:**

I have no major questions.

**Alternative Views Section:**

Yes

**Compliance With Llm Reviewing Policy A Conservative:**

Affirmed.

**Discussion Potential:**

3

**Final Justification:**

I enjoyed reading the paper overall and I think it is a good fit to be of value to the ICML community.

**Paper Summary:**

The paper argues that current claims on interpretability research are limited from a perspective of causal reasoning; instead, claims should be better framed using causal inference frameworks. The paper supports this claim with a detailed literature review, case studies, and a call to action.

**Position:**

Yes

**Position In Title:**

Yes

**Related Work:**

4

**Strengths And Weaknesses:**

Overall I found the paper well written and the position strongly argued. I do not see any major weaknesses in the work.
- The paper is supported by a set of reasoned arguments as well as concrete examples and case studies drawn from an annotated literature review to show limitations in current research claims.
- The argument is clearly presented.
- The paper also includes a series of research questions in a call to action to support the position. These questions may guide future discussion and better shape future research in this field.

**Support:**

4

---

> ### Author Rebuttal · Authors · 2026-03-31
>
> We thank you for your strong endorsement of our paper. Your assessment that *the position [is] strongly argued* with *no major weaknesses*, supported by *reasoned arguments as well as concrete examples and case studies*, is deeply appreciated. We are glad you find that the research questions may *guide future discussion*, and are happy to engage in it.

---

> > ### Author Rebuttal · Reviewer_FGv2 · 2026-04-02
> >
> > I did not have any major concerns in my initial review. Upon reading the rebuttals, my general position remains the same.

---

### Official Review · Reviewer_gyZF · 2026-03-08

**Significance:** 3
**Argument Clarity:** 2
**Rating:** 3
**Confidence:** 4

**Questions:**

Please see Strengths And Weaknesses

**Alternative Views Section:**

Yes

**Compliance With Llm Reviewing Policy A Conservative:**

Affirmed.

**Discussion Potential:**

2

**Final Justification:**

As some concerns remains, I would like to maintain my rating.

**Paper Summary:**

This paper presents a position on how causality can be used to define or formulate interpretability questions. The main point is that different interpretability claims correspond to different levels of Pearl’s hierarchy. The draft also highlights CRL as a potentially useful tool for interpretability, particularly focusing on identifiability.

**Position:**

Yes

**Position In Title:**

Yes

**Related Work:**

2

**Strengths And Weaknesses:**

The paper discusses an important and interesting topic. I agree it is necessary to formulate interpretability more rigorously, but several mains points may need to be justified.

1) The main point, using causality to formulate interpretability, is not new to me, since, I think, one of the main motivations for using causality is its ability to answer 'why', and this is already well accepted in the community. For this, I think it would help if the paper could more clearly explain what additiona insight is gained here.

2) I am not fully convinced by framing all interpretability questions under a causal perspective. My impression is that causal formalization is clearly useful for some interpretability settings, especially when interventions are involved, but whether all interpretability questions naturally fall into causal interpretability may still need to be clarified more carefully. It may therefore help to more explicitly distinguish the boundary between causal and non-causal interpretability, so that the role of causality can be discussed within a clearer scope. As a result, the Section title, 'Interpretability Questions Are Causal Questions', may need to be carefully used.

3) Also, I am not fully convinced by the role of CRL. CRL is one **specific line** within causality, and, for which there are some basic assumptions, e.g., latent generative process. These assumptions may be natural in some settings, e.g., for unstructured data, but whether they are suitable for text representations is itself still an open question. It may therefore be helpful to explain more clearly why CRL is the most relevant framework here, and if neceesary, some basic assumptions should be clearfied to provide a clear context.

4) The last concern is regarding identifiability, especfically in the context of CRL. In general, identifying causal rerepresentations often relies on certain assumptions that are difficult to test in practice, e.g., typically, multiple environmental data generated by interventions on latent mechanisms. Although identifiability can be provided theoretically under such assumptions, the validity of these assumptions is often not directly verifiable. Because of this gap, I am not fully sure how far identifiability alone can support interpretability claims in practice.

Suggestions:

It may help if the paper discusses more explicitly the distinction between the general scope of interpretability and the scope of causal interpretability (major), and also places the CRL identifiability literature in a slightly broader context (minor).

**Support:**

2

---

> ### Author Rebuttal · Authors · 2026-03-31
>
> We thank you for engaging deeply with our work. Your agreement that *it is necessary to formulate interpretability more rigorously* and that the topic is *important and interesting* is encouraging, an assessment shared by all the other reviewers as well. We address your concerns below.
>
> > W1
>
> We'd draw attention to our emphasis on *identifiability*, which goes well beyond a general awareness that causal questions matter. It is not enough to know we want to answer "why", what's crucial is knowing *what data answers those questions* and *to what degree*. The paper additionally proposes affordance-based identifiability, that does not rely on ground-truth generative factors (Sec 2.2). Our framework of rung mismatch + identification gap as a named, operationalisable diagnostic enables predicting specific failure modes. Reviewer sPgE independently validates this as well, and we empirically validate it in our pilot study. We also provide a checklist to make this diagnostic actionable.
>
> We will clarify that our contribution is not just that causality is relevant to interpretability, but rather, how we propose to make this actionable by providing identifiability-grounded diagnostics that specify what data answers a given interpretability question and to what degree.
>
> > W2
>
> Our observation is that claiming a component "mediates," "encodes," or "controls" a behaviour, is a causal claim, whether or not we use a causal formalism[1, 2]. The causal framework is general: it includes purely descriptive queries ("what tokens does this head attend to?") as well, which are L1. It also enables us to posit interventional and counterfactual queries in a unified manner. That said, we would be genuinely interested if you can point to a concrete counterexample, as it would help us refine the boundary of this claim.
>
> [1] https://arxiv.org/pdf/2301.04709
>
> [2] https://www.jstor.org/stable/10.1086/341859
>
>
> > W3
>
> We would like to respectfully push back on the premise that CRL is narrowly tied to recovery of latent generative factors. We cite recent CRL works identifying *latent mediating features* (variables that mediate the effect of input on output rather than generating observations directly) [1, 2, 3]. These references make clear that CRL's scope extends well beyond the standard "recover generative factors" setup. That said, we agree it is worth being explicit about where assumptions are more or less likely to hold and we will make this clearer in the paper.
>
> A *central goal* of the paper is not just to import CRL into mechanistic interpretability (MI), but to propose a novel direction for CRL itself in order to contribute to both fields. We will make our writing reflect this clearer, thank you for raising the point. In the paper, we propose developing CRL frameworks that treat identifiability as interaction-relative in Sec 2.2, which we open by stating identifiability explicitly as “a statement about what structure is invariantly recoverable, not about what entities exist”. We go further to acknowledge that LLMs may encode structure that humans do not conceptualise at all, requiring neologisms rather than better labels in Sec 2.2 [4, 5, 6, 7]. Together, these moves are aimed to extend standard CRL toward accommodating representations whose internal language may be fundamentally alien to human categories, just as formalising the causal commitments in MI would help benefit from that broadened machinery. We see developing both fields collaboratively as essential to reliably solving bigger problems.
>
> [1] https://arxiv.org/abs/2211.14666
>
> [2] https://arxiv.org/abs/2405.20482
>
> [3] https://openreview.net/forum?id=Zf6Oj5x9sE
>
> [4] https://arxiv.org/pdf/2310.16410
>
> [5] https://arxiv.org/abs/2510.08506
>
> [6] https://openreview.net/forum?id=O4LaRH4zSI
>
> [7] https://philsci-archive.pitt.edu/16034/1/The%20Comparative%20Psychology%20of%20Artificial%20Intelligence%204%20-%20with%20figures.pdf
>
>
> > W4
>
> We argue that identifiability is an *unavoidable* question if we want valid causal inferences. We can never avoid assumptions, but the question is whether they are implicit or explicit. Making them explicit provides clarity about good regularisers, good evaluation metrics, and when to expect failure. We agree that CRL may not always provide immediate answers, but we do claim that CRL can help us in practical ways—for example, specifying what kinds of structure are recoverable from what kinds of evidence, reducing the experimental search space. As we argue in Sec 2.2, there is no unsupervised interpretability, since supervision enters through affordances, and identifiability characterises what those affordances can reveal. In Sec 4.2: Task Relative Equivalence Classes, we also make it clear that identifiability is a spectrum, and not every interpretability task may benefit from the strongest kind of identifiability.
>
> Thank you for your suggestions on clarifying scope and the broader context of CRL, we look forward to the follow-up discussion.

---

> > ### Author Rebuttal · Reviewer_gyZF · 2026-04-03
> >
> > Thanks for the response. I appreciate the clarifications, which address part of my confusion.
> >
> > However, I still find some points unclear. In particular, the notion of identifiability in this setting remains somewhat ambiguous to me. In conventional CRL, identifiability is defined with respect to latent generative variables, once the conventional CRL is relaxed, it is not entirely clear what is being identified. Alos, the concern raised in my initial comments still remains, i.e., how the gap can be concretely characterised or addressed in practice.
> >
> > I currently maintain my rating, and will consider the perspectives of other reviewers in the subsequent discussion.

---

### Official Review · Reviewer_J2gy · 2026-03-10

**Significance:** 3
**Argument Clarity:** 3
**Rating:** 5
**Confidence:** 3

**Questions:**

- Can you expand on the difference between L2 and L3?
- Does L3 evidence automatically mean you can make statements in L2 and L1?

**Alternative Views Section:**

Yes

**Compliance With Llm Reviewing Policy A Conservative:**

Affirmed.

**Discussion Potential:**

3

**Final Justification:**

Maintaining my rating of 5. Overall, I believe this is a strong paper that addresses an important problem, is clearly written and has some actionable recommendations. My questions were partially answered in the rebuttal, though I think the authors still need to more clearly describe the difference between L2 and L3 evidence. In particular, it's not clear to me what "the effect of removing only that head while everything else behaves as it did in the original forward pass" means in their latest response, is this just the direct contribution from head to output logits, ignoring second and higher order effects?

**Paper Summary:**

This paper presents the position that interpretability claims should be clearly specified in terms of Pearl's causal ladder. They find that many existing papers are either vague in their claims, or overclaim their findings, i.e. make statements that are not supported by their evidence. For example, papers may make causal claims based on associational evidence, or the findings are not sufficiently identifiable can could also be explained by alternative hypotheses.

**Position:**

Yes

**Position In Title:**

Yes

**Related Work:**

3

**Strengths And Weaknesses:**

Strenghts:
- Addresses an important problem in interpretability and provides a usefl model for making more rigorous interpretability claims.
- Clearly written and well argumented
- Good use of examples and case-studies to show how these theoretical concepts map to existing interpretability work.
- Actionable insights with suggested future direction and a checklist for authors

Weaknesses:
- Few things could be explained more clearly, such as the difference between L2 and L3 evidence, and whether evidence at a higher rung is sufficient for making lower rung claims.
- Sometimes assumes strongest interpretation of vague claims, for example its not clear whether auto-interp results are typically thought to represent L3 knowledge.

Minor issues:
- Making case studies their own section would make the structure more clear
- Sec 3 starts as “before applying our framework to cases studies” but is presented after the case studies.

**Support:**

3

---

> ### Author Rebuttal · Authors · 2026-03-31
>
> We thank you for your encouraging review and recognition that the paper *addresses an important problem* and *provides a useful model for making more rigorous interpretability claims*, which captures our intent. We are particularly glad you found the case studies effective in showing *how these theoretical concepts map to existing interpretability work* and that the checklist provides *actionable insights with suggested future direction*. We address your suggestions below.
>
> > L2 vs L3 distinction could be clearer; does L3 evidence license L2/L1 claims?
>
> Yes, the hierarchy is cumulative upward: L3 licenses L2 and L1 (a counterfactual answer entails the interventional and associational ones). The reverse does not hold. The key distinction between L2 and L3 is that L2 asks 'what happens if I intervene?' (forward-looking, manipulating the actual system), while L3 asks 'what would have happened had things been different?' (backward-looking, reasoning about an alternative to an outcome already observed). For eg, L2 asks 'if I ablate this head, does the model still refuse?' whereas L3 asks 'given that the model refused, would it have refused had this head been inactive?'—the latter requires reasoning over a world contrary to what actually occurred. Given that we do not have access to this contrary world, we can only approximate L3 answers by constructing counterfactual scenarios through structural assumptions about the causal model where some assumptions are already implicit in several methods (as mentioned in the case studies and examples in the paper)—for instance, causal scrubbing assumes that a circuit's functional dependence structure is preserved under resampling, which is an explicit structural commitment of the kind L3 requires. This suggests that L3 is not unachievable, but closing it requires making these implicit assumptions explicit so practitioners can assess which counterfactual claims their setup actually supports.
>
> We will make our writing clearer in Sec 4.1, in the causal ladder box and the safety example.
>
>  > Sometimes assumes strongest interpretation of vague claims (e.g., auto-interp as L3)
>
> We thank the reviewer for pinpointing our imprecise phrasing. We do not assume authors intend the strongest reading. Our point is that vague language *admits* it. When "this feature represents [concept]" lacks causal bookkeeping, a downstream user may often make the stronger interpretation relying on a heuristic that is not well-grounded. The problem is not intent but imprecise language creating the *possibility* of misinterpretation. Our framework provides vocabulary to close this gap.
>
> *Proposed revision:* We will reframe Table 1 to use "Aim" and "What the Evidence Supports" as column headers (replacing "Common Interpretation" / "What Evidence Licenses"), making clear we describe what the *method aspires to*, not what authors claim. We will also clarify this in the pilot study box.
>
> > Sec 3 forward reference
>
> Thank you for pointing this out. We will replace "Before applying our framework to case studies" (line 571) with "Having applied our framework above, we now situate it relative to..."

---

> > ### Author Rebuttal · Reviewer_J2gy · 2026-04-04
> >
> > Thank you for the response!
> >
> > I am still a bit unclear on a few points.
> >
> > 1: Difference between L2 and L3 evidence:
> > - I think the distinction makes sense to if we think about a physical system, I struggle to understand how this applies to an LLM. The two questions you pose appear identical to me since we can just repeat exactly the same experiment in an LLM so the distinction between past and future doesn't seem very meaningful.  To answer  'if I ablate this head, does the model still refuse?', you can run a new experiment, ablate the head and measure if the model still refuses. To answer: 'given that the model refused, would it have refused had this head been inactive?', you ablate the head, run the model again and see if the model refuses. What is the difference?
> >
> > 2. Table 1
> >
> > I believe using "what evidence supports" is a better title, but I'm not sure if "Aim" is an improvement over Common Interpretation. This supposes that the goal of these methods is to provide level 3 evidence but I'm not fully sure if that is correct. For example, many input-based auto-interp methods are actually aiming to provide L1 evidence in my opinion, even if they don't explicitly state it.

---

### Decision · Program_Chairs · 2026-04-30

**Decision:**

Accept (regular)

**Comment:**

This paper received a range of scores. After careful consideration, the paper has enough merits for acceptance